# Human RIPK3 maintains MLKL in an inactive conformation prior to cell death by necroptosis

Yanxiang Meng [1,2,4], Katherine A. Davies [1,2,4], Cheree Fitzgibbon[1,2], Samuel N. Young[1], Sarah E. Garnish[1,2], Christopher R. Horne [1,2], Cindy Luo[1], Jean-Marc Garnier[3], Lung-Yu Liang[1,2], Angus D. Cowan [1,2], Andre L. Samson [1,2], Guillaume Lessene [1,2], Jarrod J. Sandow [1,2], Peter E. Czabotar [1,2,5✉] & James M. Murphy [1,2,5✉]

The ancestral origins of the lytic cell death mode, necroptosis, lie in host defense. However, the dysregulation of necroptosis in inflammatory diseases has led to widespread interest in targeting the pathway therapeutically. This mode of cell death is executed by the terminal effector, the MLKL pseudokinase, which is licensed to kill following phosphorylation by its upstream regulator, RIPK3 kinase. The precise molecular details underlying MLKL activation are still emerging and, intriguingly, appear to mechanistically-diverge between species. Here, we report the structure of the human RIPK3 kinase domain alone and in complex with the MLKL pseudokinase. These structures reveal how human RIPK3 structurally differs from its mouse counterpart, and how human RIPK3 maintains MLKL in an inactive conformation prior to induction of necroptosis. Residues within the RIPK3:MLKL C-lobe interface are crucial to complex assembly and necroptotic signaling in human cells, thereby rationalizing the strict species specificity governing RIPK3 activation of MLKL.

[1] Walter and Eliza Hall Institute of Medical Research, 1G Royal Parade, Parkville, VIC 3052, Australia. [2] Department of Medical Biology, University of Melbourne, Parkville, VIC 3052, Australia. [3] SYNthesis med chem, 30 Flemington Rd, Parkville, VIC 3052, Australia. [4]These authors contributed equally: Yanxiang Meng, Katherine A. Davies. [5]These authors jointly supervised this work: Peter E. Czabotar, James M. Murphy. ✉email: czabotar@wehi.edu.au; jamesm@wehi.edu.au

Necroptosis is a lytic, caspase-independent programmed cell death modality[1–6] that has ancestral origins in innate immunity[7–11]. Recent studies have implicated the dysregulation of necroptosis signaling in a range of human pathologies, including inflammatory diseases[12–15], renal injury[16,17], diabetes[18], and inflammatory bowel disease[19], raising interest in the pathway as a therapeutic target[20]. Necroptotic signaling is initiated upon ligation of death or pathogen receptors, such as tumor necrosis factor (TNF) receptor 1 or Toll-like receptors, and the subsequent assembly of a cytoplasmic signaling platform termed the necrosome. Necrosome formation occurs in scenarios where the cellular inhibitors of apoptosis (cIAP) E3 Ubiquitin ligase family or Caspase-8 protease are disabled, preventing the involvement of the apical kinase, receptor-interacting serine/threonine protein kinase (RIPK)-1, in nuclear factor-κB (NF-κB) or apoptotic signaling. The necrosome is a high molecular weight complex in which the RIPK1 and RIPK3 kinases assemble as core proteins via their C-terminal functional amyloid-forming RIP homotypic interaction motifs (RHIMs)[21–23]. Although the details of the underlying cues for assembly are incompletely understood, necroptotic signaling induces autophosphorylation of the RIPK1 kinase domain, which permits recruitment and auto-activation of RIPK3 (refs. [5,24,25]) and subsequent engagement of the terminal pathway effector, the Mixed Lineage Kinase domain-Like (MLKL) pseudokinase[4,6,26,27]. RIPK3-mediated phosphorylation of MLKL permits its dissociation from the necrosome, prior to trafficking to the plasma membrane where MLKL accumulates into hotspots and permeabilizes the cell, manifesting in a lytic, pro-inflammatory form of death[28–37].

Although the core principles of necroptotic signaling are conserved between mammals, there is remarkable species specificity[38,39], such that rat MLKL cannot substitute for mouse MLKL in mouse cells despite their 86% amino acid identity[40]. Such divergence is consistent with an ancestral origin for the pathway in host defense, where exposure to necroptosis-interfering proteins encoded by different pathogens may have driven the co-evolution, and interspecies divergence, of RIPK3 and MLKL cognate pairs[7–11,41–44]. The precise nature of the interactions between RIPK3 and MLKL are of immense interest, because recognition of MLKL is a key checkpoint in the execution of the necroptosis pathway. To date, our understanding of this interaction has been limited by a dearth of RIPK3:MLKL complex structures, with only the mouse complex reported, and by the availability of only the mouse RIPK3 structure[45,46]. Recent data indicate that human MLKL occurs in the cytoplasm in complex with RIPK3 prior to initiation of necroptosis[47]. Upon receipt of necroptotic signals, this complex is recruited to the necrosome where RIPK3 phosphorylates and activates MLKL. Phosphorylation was proposed to induce MLKL release from the necrosome, which could be detected by a recently-described synthetic protein ligand termed Monobody-27 (ref. [47]). The Monobody-27 recognition site on human MLKL was proposed to overlap the RIPK3-binding site on MLKL, because Monobody-27 binding was occluded prior to MLKL activation and necrosome disengagement. Additionally, this process was proposed to involve a structural interconversion of MLKL from an open to a closed conformation[47]. However, which form represents the dormant form bound by RIPK3, and which is the activated form following RIPK3-mediated phosphorylation, required validation. Further, owing to primary sequence divergence between mouse and human MLKL and RIPK3, knowledge of the RIPK3-binding site on MLKL has relied on the identification of synthetic binding proteins, termed monobodies, that compete with RIPK3 for binding to MLKL[47]. However, a precise definition of how human RIPK3 recognizes human MLKL, and how this differs from the mouse counterpart, has been lacking in the absence of a complex structure.

Here, we report the crystal structure of the human RIPK3 kinase domain alone and in complex with the human MLKL pseudokinase domain. Human RIPK3 and MLKL bound in a face-to-face arrangement mediated by parallel N-lobe:N-lobe and C-lobe:C-lobe contacts, which lead to their activation loops and ATP-binding sites facing towards each other. Interestingly, despite only 70% sequence identity and different autophosphorylation sites between human and mouse RIPK3 kinase domains, the C-lobes of human and mouse RIPK3 exhibited comparable topology and modes of interaction with MLKL, while the N-lobes diverged structurally and formed a distinct interface in the human RIPK3:MLKL complex. These differences could be attributed to human MLKL assuming an open conformation with a helical activation loop that displaces the key regulatory element, the αC helix. In contrast, in the orthologous mouse RIPK3:MLKL complex structure, the mouse MLKL activation loop forms a shorter helix that packs against the MLKL C-lobe in a distinct orientation. The different positions of the human and mouse MLKL activation loop helices impart distinct modes of interaction with the N-lobe of human and mouse RIPK3 in their respective complexes. However, it is the interactions between the RIPK3:MLKL C-lobes that proved crucial for complex formation and necroptotic signaling in human cells. Extensive mutational analysis identified a bidentate interaction as critical for necroptotic signaling, where phospho-S227 of human RIPK3 engages in electrostatic interactions with MLKL C-lobe residues, and hydrophobic residues centered on F386 in the MLKL C-lobe project into a hydrophobic cavity in the RIPK3 C-lobe. The critical functions of these residues in human RIPK3 and MLKL pinpoint the interface between their C-lobes as the crucial determinant of RIPK3:MLKL selective recognition across species.

## Results

**Phosphorylated human RIPK3 complexes with unphosphorylated human MLKL in crystallo.** To dissect the nature of the interaction between human RIPK3 and MLKL, and how RIPK3 activates MLKL, we co-expressed and purified the human RIPK3 kinase domain (residues 1–316) and the MLKL pseudokinase domain (residues 190–471) complex from Sf21 insect cells (hereafter referred to as human RIPK3:MLKL). Similar to the previously reported mouse counterpart[46], recombinant human RIPK3 kinase domain and MLKL pseudokinase domain formed a stable complex when co-expressed. To aid crystallization, we introduced the C3S and C110A mutations into human RIPK3, because previous studies with mouse RIPK3 identified these unpaired cysteines as a source of heterogeneity due to interchain disulfide bond formation[46]. Initial sparse matrix crystallization screens failed to afford crystals of this binary complex, likely due to the intrinsic instability of human RIPK3 (Supp. Fig. 1a), which begins to unfold and therefore rapidly precipitates at room temperature. We could, however, generate diffracting complex crystals upon stabilizing RIPK3 by inclusion of the previously reported RIPK3 inhibitor, BMS compound 10 (ref. [48]; hereafter referred to as Compound 10), in our crystallization screens.

We determined the human RIPK3:MLKL co-crystal structure with Compound 10 bound to 2.23 Å resolution (Table 1 and Fig. 1a, b), with one copy of each protein in the asymmetric unit in the $P2_1 2_1 2_1$ space group. The overall mode of interaction between human RIPK3 and MLKL grossly resembles that observed in the mouse RIPK3:MLKL co-crystal structure (PDB: 4M69 (ref. [46]); RMSD = 0.699 Å over 201 Cα atoms; Fig. 1c–e). Human RIPK3 and MLKL form a 1:1 complex in a face-to-face orientation, with interfaces between their respective N-lobes and C-lobes, and their ATP-binding pockets facing each other, offset by an angle of 108.2°.

**Table 1 X-ray crystallography data collection and refinement statistics.**

| Structural parameters | Human RIPK3 (residues 1–316; C3S, C110A): human MLKL (residues 190–471) complex (PDB: 7MON) | Human RIPK3 (residues 2–316; C3S, C110A) with GSK´843 (PDB: 7MX3) |
|---|---|---|
| **Data collection statistics** | | |
| Wavelength (Å) | 0.9537 | 0.9537 |
| Resolution range (Å) | 42.42–2.23 (2.31–2.23)[a] | 80.76–3.23 (3.67–3.23) |
| Space group | $P\,2_1\,2_1\,2_1$ | $P\,2_1\,2_1\,2_1$ |
| Unit cell (x, y, z, α, β, γ) | 64.495 82.839 104.842 90 90 90 | 104.82 104.82 126.68 90 90 90 |
| | | Spherical[b] \| elliptical truncation[c] |
| Total reflections | 186,578 (18125) | 147,324 (7411) \| 67,513 (3021) |
| Unique reflections | 27,979 (2716) | 23,021 (1143) \| 10,790 (540) |
| Multiplicity | 6.7 (6.7) | 6.4 (6.5) \| 6.3 (5.6) |
| Completeness (spherical) (%) | 99.71 (98.69) | 100 (100) \| 46.9 (7.2) |
| Completeness (ellipsoidal) (%) | – | − \| 90.7 (56.9) |
| Mean I/sigma (I) | 11.26 (1.38) | 1.8 (0.2) \| 3.4 (1.5) |
| Wilson B-factor | 36.18 | − \| 63.2 |
| R-merge | 0.1206 (1.347) | 0.581 (12.309) \| 0.259 (1.031) |
| R-meas | 0.1309 (1.46) | 0.634 (13.388) \| 0.283 (1.142) |
| R-pim | 0.05035 (0.5567) | 0.250 (5.218) \| 0.113 (0.484) |
| CC1/2 | 0.998 (0.564) | 0.970 (0.076) \| 0.979 (0.451) |
| **Refinement statistics** | | |
| Refinement resolution range (Å) | 41.42–2.23 (2.28–2.23) | 74.12–3.23 (3.37–3.23) |
| Reflections used in refinement | 27,968 (2715) | 10,786 (49) |
| Reflections used for R-free | 1998 (194) | 1093 (6) |
| R-work | 0.2088 (0.2935) | 0.2394 (0.4527) |
| R-free | 0.2600 (0.3395) | 0.2855 (0.3577) |
| Number of non-hydrogen atoms | 4544 | 8956 |
| Macromolecules | 4365 | 8839 |
| Ligands | 74 | 108 |
| Solvent | 105 | 9 |
| Protein residues | 553 | 1129 |
| RMS (bonds) | 0.002 | 0.005 |
| RMS (angles) | 0.53 | 0.93 |
| Ramachandran favored (%) | 97.74 | 95.04 |
| Ramachandran allowed (%) | 2.26 | 4.96 |
| Ramachandran outliers (%) | 0 | 0 |
| Rotamer outliers (%) | 1.87 | 0.61 |
| Clashscore | 2.6 | 10.94 |
| Average B-factor | 43.63 | 67.32 |
| Macromolecules | 43.6 | 67.36 |
| Ligands | 51.41 | 65.15 |
| Solvent | 39.49 | 47.98 |

[a]Statistics for the highest-resolution shell are shown in parentheses.
[b]Statistics are shown for spherical processing of the diffraction data to 3.23 Å in autoPROC with STARANISO.
[c]Statistics are shown for elliptical truncation of the diffraction data to 3.23 Å in autoPROC with STARANISO, which was performed owing to the severe anisotropy within the RIPK3:GSK'843 dataset.

Consistent with earlier studies implicating human RIPK3 S227 phosphorylation as a determinant of MLKL interaction[4,47,49], S227 is phosphorylated in the human RIPK3:MLKL co-crystal structure with clear electron density for the phosphate group (Supp. Fig. 1b). Similarly, T224 of RIPK3 is also phosphorylated in the co-crystal structure (Supp. Fig. 1b), raising the possibility that this residue may perform a role in augmenting MLKL binding and conferring species selectivity. The previously characterized human MLKL activation loop phosphosites, T357 and S358, are both modeled with clear sidechain electron density; however, neither of these residues are phosphorylated in our structure (Supp. Fig. 1c), consistent with the idea that human MLKL phosphorylation by RIPK3 drives its dissociation from RIPK3 (ref. [47]).

Compound 10 was observed in the ATP-binding pocket of human RIPK3 and also, unexpectedly, in that of MLKL (Fig. 1a, b and Supp. Fig. 1d, e). Within human RIPK3, the compound forms extensive contacts with the N-lobe β-sheets, αC helix, and the C-lobe hinge region. Compound 10 was initially described as a Type II kinase inhibitor; however, our structure reveals that the compound's 4-fluorophenyl tail group fails to insert deeply into the back pocket of RIPK3 and flip out the DFG motif in a classic Type II inhibitor mode, unlike BMS Compound 9 bound to mouse RIPK3 (PDB: 6OKO)[48]. Instead, the 4-fluorophenyl moiety occupies the space between the αC helix (residues 57–66) and β4 strand, interrupting the conserved salt bridge between K50 and E60 (Fig. 1b). We also observed density in the ATP-binding pocket of human MLKL where half of Compound 10, from the cyclopropyl to the 3-fluoro-4-alkoxyanilide moiety, occupied the ATP-binding site with half of the compound solvent exposed (Supp. Fig. 1e and Fig. 2a). This binding mode is distinct from that observed in RIPK3, with binding centered on the β5-αD loop of human MLKL (Supp. Fig. 1e), thus reflecting differences in geometry between the human RIPK3 and MLKL ATP-binding sites. Consistent with this unusual binding mode, we observed only a modest thermal shift for human MLKL pseudokinase domain in the presence of 40 μM Compound 10 using differential scanning fluorimetry (Supp. Fig. 1a), supporting the propensity of

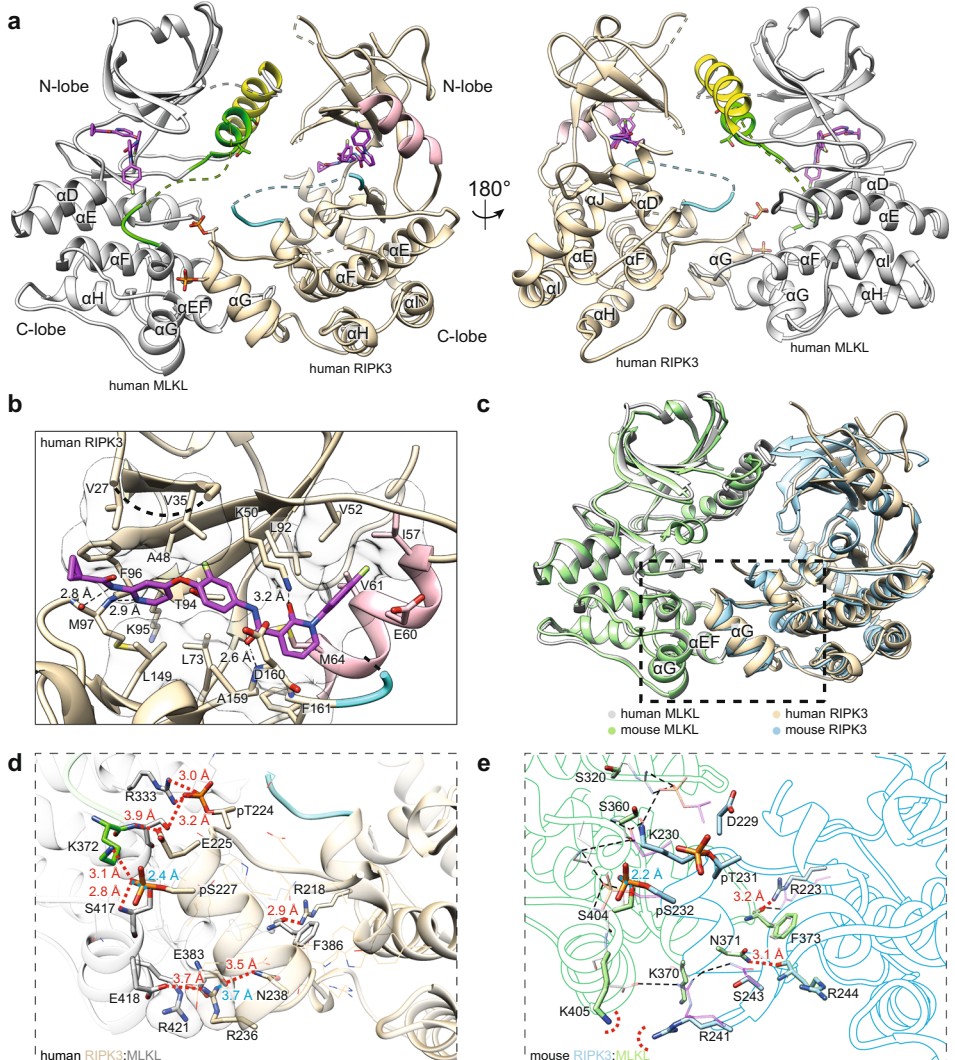

**Fig. 1 The human MLKL pseudokinase domain and RIPK3 kinase domain form a face-to-face complex. a** Orthogonal views of the co-crystal structure of human RIPK3 (residues 1–316; C3S, C110A):human MLKL (residues 190–471). MLKL (gray), αC helix (yellow), and activation loop (green) are highlighted. RIPK3 (wheat), αC helix (pink), and activation loop (cyan) are highlighted. Compound 10 (magenta) and key phosphorylation sites (RIPK3:pS227, pT224; MLKL:T357, S358), and F386 are shown as sticks. Dashed lines represent chain connectivity where insufficient electron density was observed to allow model building. **b** Zoomed view of Compound 10 binding to the human RIPK3 active site. Sidechains and backbone atoms interacting with Compound 10 (magenta) are shown as sticks. Electrostatic interactions are shown as black dashed lines. Transparent surfaces are shown for residues that hydrophobically interact with Compound 10. **c** Overlay of human RIPK3 (gray):MLKL (wheat) (this study; PDB: 7MON) with the mouse RIPK3 (green):MLKL (blue) complex (PDB: 4M69)[46]. Alignment was performed by superimposing the respective MLKL in each complex model using UCSF Chimera Matchmaker. **d** Zoomed view of the C-lobe electrostatic and hydrophobic interactions within the human RIPK3:MLKL structure. Sidechains involved in electrostatic interactions are shown as sticks; sidechains mediating non-polar interactions are shown as thin lines. Electrostatic interactions are shown with red dashed lines or in blue when hidden behind sidechains. **e** Zoomed views of the C-lobe electrostatic interactions from the mouse RIPK3:MLKL structure. Peptide backbones are shown as silhouettes. Sidechains involved in electrostatic interactions are shown as sticks and labeled. Electrostatic interactions are shown with red dashed lines or in blue when hidden behind sidechains. Dashed red semi-circles illustrate the potential electrostatic repulsion between mouse MLKL K405 and RIPK3 R241. Electrostatic interactions from the human RIPK3:MLKL structure are overlaid (thin black dashed lines) with human residue sidechains shown as thin transparent sticks.

human MLKL to bind the compound when in excess concentration, albeit weakly.

**The human RIPK3:MLKL complex in solution resembles the crystal structure.** We next sought to examine whether our complex crystal structure resembled the conformation of the RIPK3:MLKL complex in solution using small angle X-ray scattering (SAXS; statistics in Supp. Table 1). Using an inline size exclusion chromatography setup with synchrotron radiation, we examined solution scattering of human RIPK3:MLKL in the absence of, or in complex with, Compound 10 (Supp. Fig. 1f, g). In both cases, Guinier plot linearity indicated that samples were monodispersed and free of apparent aggregation or interparticle interference (Supp. Fig. 1f). Importantly, the presence of Compound 10 did not measurably impact the complex conformation, such that the scatter plots in the absence or presence of Compound 10 are almost identical ($\chi^2 = 0.166$, $p$ value = 1) with comparable radii of gyration ($R_g$) (Supp. Table 1) and maximum dimension ($D_{max}$) (Supp. Fig. 1g). Accordingly, the theoretical scattering profile based on the human RIPK3:MLKL co-crystal structure was consistent with both experimental scattering

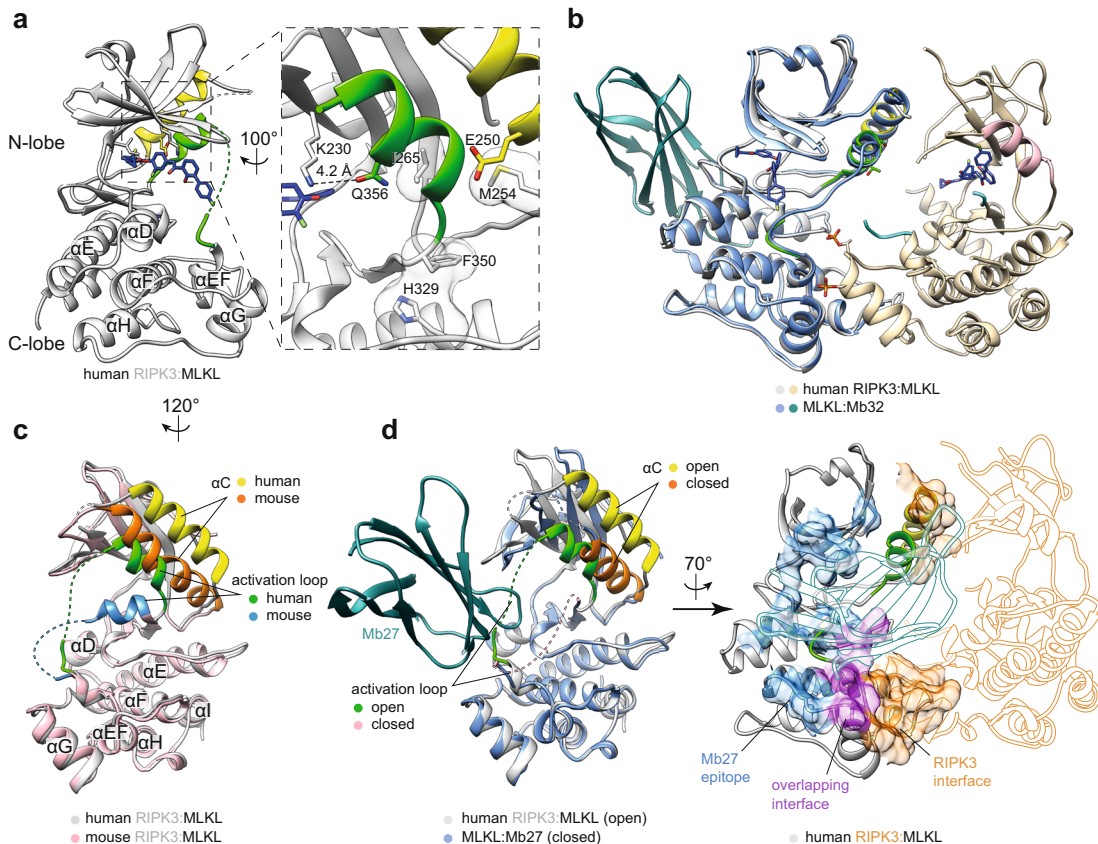

**Fig. 2 Human MLKL adopts an open conformation in complex with RIPK3. a** Human MLKL pseudokinase domain (gray) from the co-crystallized complex with RIPK3. The αC helix (yellow), activation loop (green), and Compound 10 (blue sticks) are highlighted. Zoomed view on the right illustrates the open, inactive conformation adopted by MLKL. R-spine residues (gray sticks with transparent surfaces) are misaligned; the K230–E250 salt bridge interaction is disrupted by the activation loop helix supplanting the αC helix, and contributing Q356 to the interaction with K230. **b** Superimposition of the human MLKL (blue):Mb32 (teal) complex (PDB: 7JXU)[47] with the MLKL (gray):RIPK3 (wheat) complex reported herein (PDB: 7MON). **c** Superimposition of the human (gray) and mouse (pink) MLKL structures from their complexes with RIPK3 from superimposing the respective MLKL protein in each complex model. Human complex (this study; PDB: 7MON); mouse, (PDB: 4M69)[46]. The αC helix (human, yellow; mouse, orange) and activation loop (human, green; mouse, blue) are highlighted. Viewing angle is rotated by 120° clockwise around the y-axis from the position in panel **a**. **d** A monobody (Mb27) that binds human MLKL post-release from the necrosome (left, teal; PDB: 7JW7)[47] binds MLKL via a site that overlaps the human RIPK3-binding site (right; MLKL, gray; RIPK3, wheat). The MLKL αC helix (yellow) and activation loop (green) are highlighted. Left, superimposition of the human MLKL (blue):Mb27 (teal) complex with the MLKL:RIPK3 complex (same viewing angle as panel **c**). Right, the binding epitopes on human MLKL are shown as surfaces in the cartoon (same viewing angle as panel **b**); the Mb27 binding epitope is shown in blue, the RIPK3 epitope in orange and the overlapping interface in magenta. Mb27 and RIPK3 models are shown as silhouettes. All superimpositions were performed using UCSF Chimera Matchmaker.

profiles (Supp. Fig. 1f, $\chi^2 = 0.614$ and 0.661 for apo and Compound 10-bound complexes, respectively), indicating that the heterodimer present in the crystal structure is representative of the biological assembly in solution.

**RIPK3-bound human MLKL adopts an inactive open conformation.** The human MLKL pseudokinase domain has been observed predominantly in a closed, active-like conformation in structures reported to date[46,50–52], which was recently proposed to represent the activated form of MLKL[47]. In contrast, in the human RIPK3-bound complex, the human MLKL pseudokinase domain adopts an open, inactive kinase-like conformation (Fig. 2a). This conformer is comparable to that previously reported to be stabilized by a hinge-binding monobody ligand, Mb32 (ref. [47]) (PDB: 7JXU; RMSD = 0.655 Å over 261 Cα atoms; Fig. 2b), which argues against a role for Compound 10 in promoting an open MLKL conformation in the human RIPK3:MLKL complex structure. The open conformer of human MLKL is characterized by the activation loop forming a helix antiparallel to the αC helix, which is displaced. As a result, the ATP-positioning

K230 in the β3-strand of the N-lobe is proximal to Q356 in the activation loop helix, rather than forming a salt bridge with E250 in the αC helix (Fig. 2a). Interestingly, unlike the counterpart residues in mouse MLKL, mutation of K230 or Q356 did not promote human MLKL activation[47,51], consistent with the notion that human MLKL is more tightly regulated and less prone to mutational activation than its mouse ortholog. The disruption of this salt bridge and hydrophobic regulatory (R)-spine interactions[53] observed in our human MLKL structure within this complex are hallmarks of an inactive kinase conformation. It is notable that in its complex with mouse RIPK3, mouse MLKL also adopts an open conformation (Figs. 1c and 2c and Supp. Fig. 3a, b), although the positions of the αC helix and the activation loop are markedly different (Fig. 2c). In complexed mouse MLKL, the activation loop helix is orthogonal to the αC helix, rather than antiparallel and, by virtue of contacts with the RIPK3 N-lobe loops, the αC helix is extended relative to the apo structure. Consistent with recent studies, the RIPK3-binding site on human MLKL overlaps the site recognized by the Monobody, Mb27, which is only exposed following activation and release of MLKL from the necrosome (Fig. 2d; PDB: 7JW7)[47].

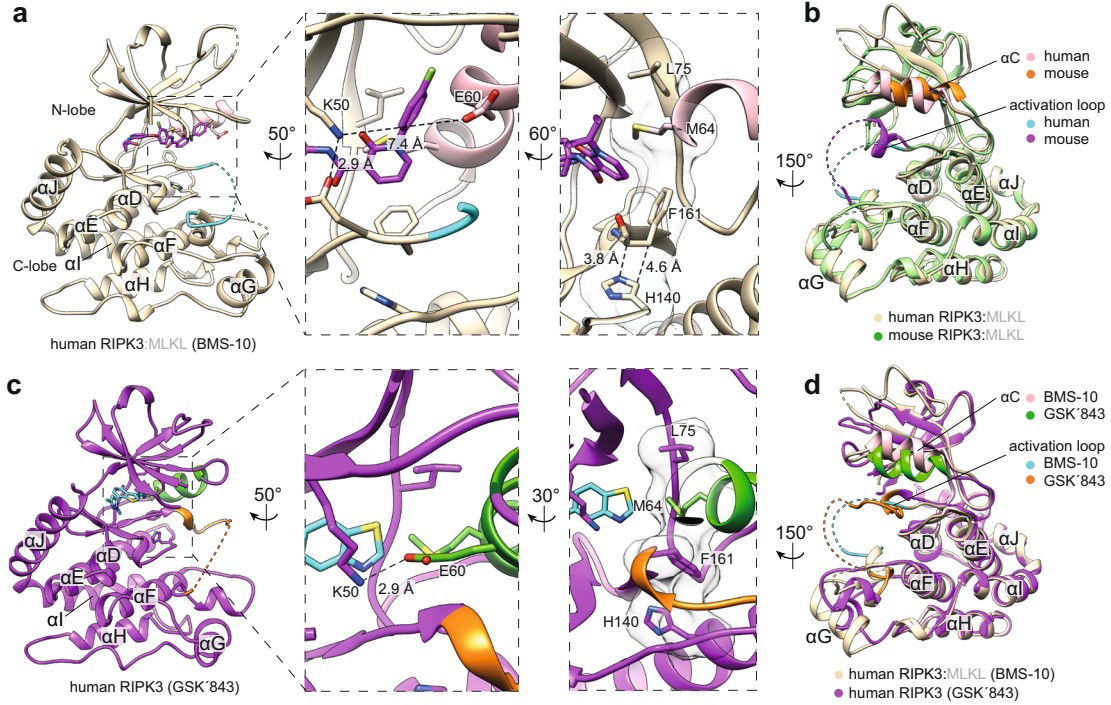

**Fig. 3 Human RIPK3 exhibits conformational plasticity. a** Human RIPK3 (wheat) from the complex with MLKL adopts an inactive conformation. Zoomed panels highlight the disrupted K50-E60 salt bridge (shown as sticks; center panel) and partial alignment of R-spine residues (shown as sticks and transparent surfaces; right panel). The αC helix (pink), activation loop (cyan), and Compound 10 (magenta sticks) are highlighted. **b** Superimposition of human (wheat; this study) and mouse (blue; PDB: 4M69)[46] RIPK3 kinase domains from their respective complexes with MLKL. Electron density was not observed for part of the activation loop, which is signified by a dashed line. **c** Structure of human RIPK3 bound to the Type I inhibitor, GSK'843 (this study; PDB: 7MX3). The K50-E60 salt bridge interaction is shown in the middle panel as a dashed line. The R-spine residues are shown in the right panel as sticks and a surface. The αC helix (green), activation loop (orange), and GSK´843 (cyan sticks) are highlighted. **d** Superimposition of human RIPK3 from the MLKL complex (PDB: 7MON; wheat) and bound to GSK'843 (PDB: 7MX3; green).

Interesting, while the C-lobe structure of RIPK3-bound human MLKL closely resembles that of the closed, active-like conformer, the positions of the N-lobes and activation loops are drastically different (Fig. 2d, left). Consistent with the open conformer serving as the RIPK3-binding form of MLKL, the N-lobe and αC helix are engaged in RIPK3-binding (Fig. 2d, right). Such interactions would not be feasible if the closed, active-like conformer of MLKL were bound to RIPK3 because the MLKL αC helix would not be poised for RIPK3 interaction. Additionally, the MLKL activation loop does not form a helix in the closed form, and the loop as modeled in the apo human MLKL structure (PDB: 4MWI)[50] would impose clashes with the human RIPK3 C-lobe. The closed conformation adopted by the human MLKL pseudokinase domain when not bound to human RIPK3 conformation likely represents the activated form of MLKL, as recently postulated[47]. While the precise choreography is incompletely understood, it would be anticipated that such an MLKL structural interconversion could dissociate the RIPK3:MLKL complex, or could arise following dissociation of the RIPK3:MLKL complex driven by MLKL phosphorylation.

**MLKL-bound human RIPK3 adopts an inactive conformation.** Human RIPK3 in complex with MLKL maintained an intact R-spine, and its activation loop was unstructured (Fig. 3a). This contrasts with the mouse RIPK3:MLKL complex, in which the R-spine of mouse RIPK3 is highly disrupted, and the activation loop forms a short helix. Like in the mouse complex structure, in our structure of human RIPK3 complexed with MLKL the conserved salt bridge between K50 of the VAIK motif and E60 of the αC helix, a hallmark of active protein kinase conformation, is

disrupted due to displacement of the αC helix (Fig. 3a). Remarkably, despite substantial sequence divergence (Supp. Figs. 2a, b and 3c, d), the C-lobe of human RIPK3 (residues 97–316) adopts an almost identical conformation to that of mouse RIPK3 (Fig. 3b; residues 98–321, RMSD = 0.658 Å over 184 Cα atoms) from the mouse RIPK3:MLKL complex structure[46]. Moreover, many of the C-lobe residues that interface with MLKL are spatially conserved or semi-conserved between human and mouse RIPK3 orthologs (Supp. Fig. 2b), although substantial sequence divergence occurs on and around phosphorylation sites. Notably, S227 in human RIPK3 occupies a position akin to that of S232 in mouse RIPK3 and, while these phosphosites are prerequisites for interaction with cognate MLKL orthologs, pS227 is shifted downwards by 2.0 Å in the human structure (Fig. 1c–e and Supp. Fig. 3e) likely because the adjacent E225 repels pS227.

The N-lobe of human RIPK3 (residues 1–97) adopts a markedly different conformation from the mouse counterpart (Fig. 3b; Supp. Fig. 3c; RMSD = 0.791 Å over 53 Cα atoms). The β1–β2, β4–β5, and β3–αC loops, and the loop preceding β1, all adopt conformations distinct from those in mouse RIPK3 (Fig. 3a, b and Supp. Fig. 3c), and each contributes to MLKL binding via an N-lobe interface. Whether the disposition of these loops and the open conformation of RIPK3 arise from the MLKL interaction remained of enormous interest. To investigate this, we next determined the structure of human RIPK3 kinase domain (residues 2–316; C3S, C110A) in the absence of MLKL, bound to the RIPK3 inhibitor, GSK'843 (ref. [54]), by X-ray crystallography to 3.2 Å resolution (Fig. 3c). The crystal lattice displayed $P2_1 2_1 2_1$ symmetry with four copies of human RIPK3 in the asymmetric unit, arranged as two disulfide linked dimers (Supp. Fig. 4a). The

disulfide bond that occurs between C234 on the αG helix of two human RIPK3 protomers arose during crystallization, because the protein subjected to crystallization trials was eluted from size exclusion chromatography as a monomer, but no reducing agent was included in crystallization conditions. We cannot exclude the possibility that this interprotomer disulfide bond could influence the disposition of the αG helix, although knowledge of the extent of any influence awaits further structural studies. The diffraction data displayed anisotropy and the best structure solution was determined following processing with autoPROC, which incorporated elliptical truncation of the data via STARANISO. Density for GSK′843 was present in all four chains in the asymmetric unit, and because we observed the best electron density for chains A and B rather than chains C and D (Supp. Fig. 4b), we have focused our analyses on the chain A structure herein.

In the absence of MLKL, and bound to the Type I inhibitor GSK′843, human RIPK3 adopted a closed, active-like conformation with an intact R-spine and K50-E60 salt bridge (Fig. 3c, electron density in Supp. Fig. 4). This confirmed that human RIPK3 in complex with human MLKL adopts an inactive conformation, with the αC helix substantially displaced from its active position (Fig. 3d). These observations parallel the mouse RIPK3 structure, which adopts an inactive conformation in complex with MLKL (Fig. 3b; PDB: 4M69)[46], while mouse RIPK3 alone (PDB: 4M66) adopts an active conformation[46]. Like in mouse RIPK3, the most substantial structural changes upon MLKL complexation occur on and around the αG helix (Fig. 3d), where autophosphorylation appears to play an important role in MLKL recognition (Fig. 1e and Supp. Fig. 3e). Accordingly, we reasoned that the inactive conformers of human and mouse RIPK3 are likely stabilized by their association with MLKL. Whether a structural interconversion to an active RIPK3 conformer is necessary to dissociate the RIPK3:MLKL complex, as proposed for MLKL[47], or whether dissociation relies entirely on RIPK3-mediated phosphorylation prompting interconversion of MLKL remain open questions.

**The C-lobes are the principal sites of human RIPK3:MLKL interaction.** The overall mode of interaction between RIPK3 and MLKL are grossly similar between mouse and human complexes, although the N-lobe positions and electrostatic interactions between C-lobes differ markedly between species (Fig. 1c–e and Supp. Fig. 3). Of the 1100 Å$^2$ estimated buried surface area within the human RIPK3:MLKL complex, the C-lobes of human RIPK3 and MLKL are the primary interaction sites (763.7 Å$^2$; $\Delta^i G = -11.4$ kcal/mol; $\Delta^i G$ $p$ value = 0.107; 10 hydrogen bonds; from PDBePISA[55]). The αE-αF loop and αG helix of human MLKL mediate the majority of the C-lobe interactions (Figs. 1a, d and 4a, b). The sidechain of F386 in the human MLKL αE-αF loop is anchored inside a hydrophobic pocket formed between the αH helix, and the αG-αH and αH-αI loops of the RIPK3 C-lobe (Fig. 4b), consistent with a previously proposed essential role for F386 of human MLKL in RIPK3 binding and necroptosis signaling[11]. The F386 hydrophobic contact appears to be augmented by the adjacent V385 of human MLKL, which are collectively surrounded by RIPK3 C-lobe hydrophobic residues, including Y205, I209, W212, V220, L222, P223, L228, V229, A232, V233, and P240 (Fig. 4b). On the periphery, a network of electrostatic contacts centered around the αG helices of both proteins stabilizes C-lobe interactions between human MLKL R333, K372, S373, S417, E418 and RIPK3 pT224, E225, pS227, R236, N238, respectively (Fig. 1d and Supp. Fig. 3c, d). Importantly, both phosphorylated residues of RIPK3 are directly involved in salt bridges at the interface, with pS227 and pT224 of RIPK3 forming salt bridges with K372 and R333 of MLKL,

respectively (Fig. 1d). The interactions of these phosphorylated residues appear integral to MLKL binding, thus providing a mechanistic explanation for the biological significance of S227 phosphorylation[4,25,47], although the precise contribution of T224 phosphorylation remains unclear.

The N-lobes of RIPK3:MLKL form a lesser interface (337.6 Å$^2$; $\Delta^i G = -2.6$ kcal/mol; $\Delta^i G$ $p$ value = 0.422; two hydrogen bonds; Fig. 4c). The activation loop helix and αC helix of MLKL interact with the β1 and β2 strands, and the β1–β2, β3–αC, and β4–β5 loops of RIPK3. This interface is stabilized by fewer interacting residues, including only one salt bridge between E25 of RIPK3 and K249 of MLKL, and a small hydrophobic patch between L26, F36, I51 of RIPK3 and I242 of MLKL. It is notable that T357 and S358 in the MLKL activation loop, which are known RIPK3 substrates, are in close proximity to this interface, and show no evidence of phosphorylation in our electron density maps (Supp. Fig. 1c). This observation supports the idea that unphosphorylated MLKL exists in complex with RIPK3 in the basal, pre-necroptotic state, and RIPK3-mediated phosphorylation of MLKL promotes dissociation from RIPK3 (refs. [11,47,51]). Consistent with this complex structure representing an inactive, basal state, the RIPK3 substrate residues, human MLKL T357 and S358, are distal from the catalytic HRD motif of RIPK3 (Supp. Fig. 3f), and a conformational change would be required for their phosphorylation to occur.

**RIPK3 S227 autophosphorylation is required for MLKL:RIPK3 interaction.** Previously, we have shown that phosphorylation of MLKL, presumably on T357 and S358, by RIPK3 would cause the RIPK3:MLKL complex to dissociate[47]. Consistent with this idea, MLKL T357 and S358 phosphorylation was not observed in the present complex structure. We next sought to determine the role of RIPK3 S227 phosphorylation on RIPK3:MLKL complex formation. We introduced either the phospho-ablating S227A mutation or kinase-inactive D142N mutation into the human RIPK3 kinase domain and co-expressed these proteins with the human MLKL pseudokinase domain in *Sf*21 insect cells. Either mutation in RIPK3, each of which eliminates S227 phosphorylation, completely abolished formation of complexes with MLKL (Fig. 5a). While both MLKL and RIPK3 were detected in the lysate, RIPK3 did not co-elute with His$_6$-MLKL pseudokinase domain following Ni-NTA pulldown and size exclusion chromatography. These data confirm pS227 on human RIPK3 is a necessary prerequisite for RIPK3:MLKL complex formation.

**Disruption of the RIPK3:MLKL C-lobe interaction prevents necroptosis.** We next sought to examine the contributions of human RIPK3:MLKL interface residues to necroptotic signaling in HT29 cells. To this end, we stably introduced doxycycline-inducible wild type or interface mutant MLKL constructs via lentiviral transduction of *MLKL*$^{-/-}$ HT29 cells, and wild type or interface mutant RIPK3 constructs into *RIPK3*$^{-/-}$ HT29 cells (Supp. Fig. 5a, b). Following doxycycline induction of exogene expression, cells were treated with a necroptotic stimulus, TSI (T, TNF; S, Smac mimetic Compound A; I, pan-Caspase inhibitor IDN-6556/Emricasan), and cell death monitored by SYTOX Green uptake using IncuCyte live cell imaging (20 h timepoint: Fig. 5b, c; 24 h timecourse: Supp. Fig. 5c, d). Strikingly, the introduction of mutations to disrupt the N-lobe interaction between RIPK3 and MLKL did not compromise necroptotic signaling. Expression of E25A, L26R, or F36W RIPK3 mutants in *RIPK3*$^{-/-}$ HT29 cells or S239D, I242A, Q245A, or K249A MLKL mutants in *MLKL*$^{-/-}$ HT29 cells enabled reconstitution of the necroptosis pathway, with cell death kinetics and magnitude comparable to their wild-type counterparts (Fig. 5b, c and Supp.

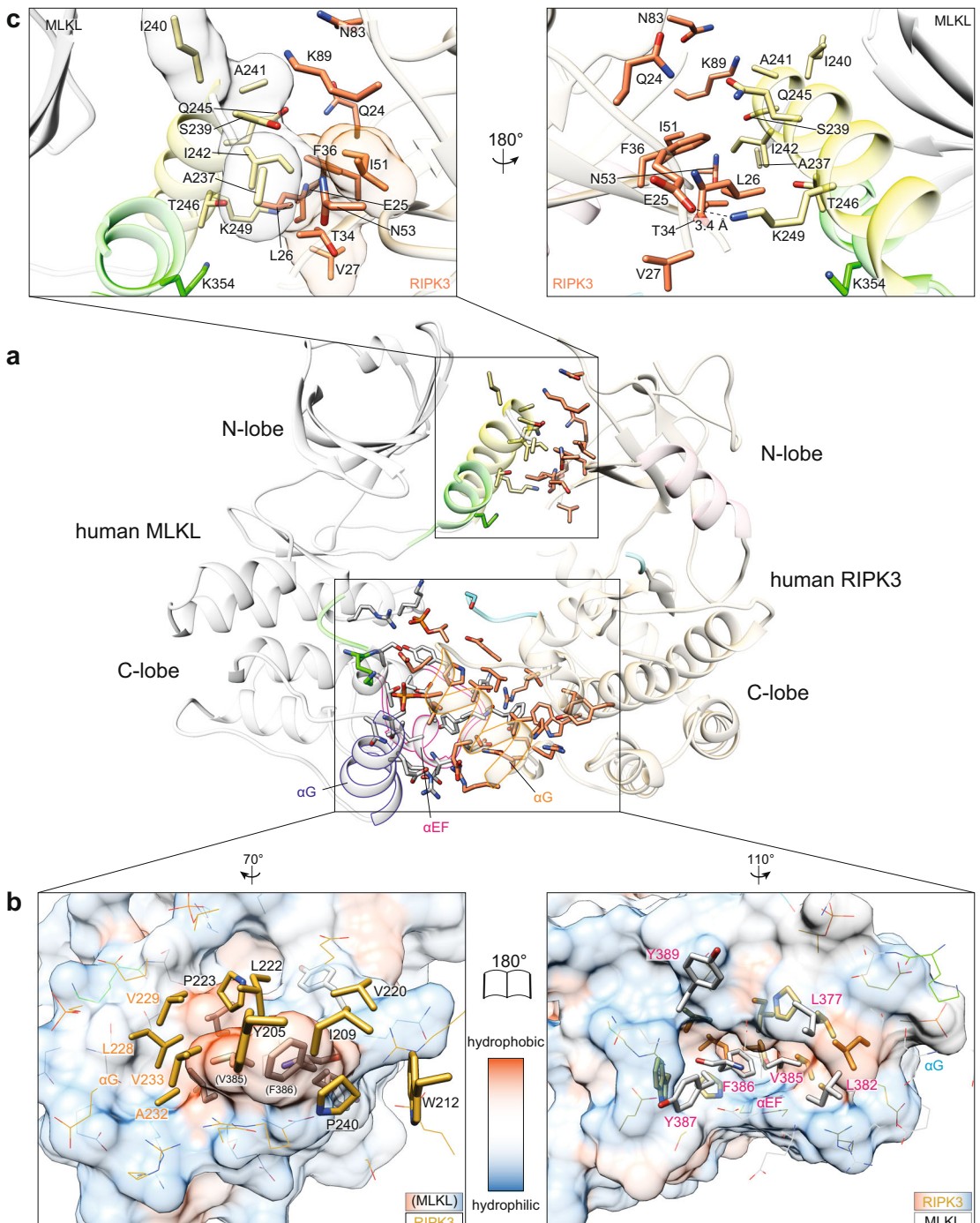

**Fig. 4 The human RIPK3 and MLKL complex interface is dominated by C-lobe interactions. a** The N-lobe interaction interface (top box) is less extensive than the C-lobe interface (bottom box). MLKL is colored gray with αC helix (yellow) and activation loop (green) highlighted. RIPK3 is colored magenta with αC helix (wheat) and activation loop (cyan) highlighted. Sidechains of interface residues (as predicted by PDBePISA) are shown as sticks. **b** Open book view of hydrophobic interacting residues within the C-lobe interface. RIPK3 sidechains are shown as orange sticks and MLKL sidechains as gray. The surfaces (left, MLKL; right, RIPK3) are colored according to hydrophobicity (red) and hydrophilicity (blue), as calculated by UCSF Chimera[76]. Other polar interface residues are shown as lines. **c** Zoomed and orthogonal views of the N-lobe interface of human RIPK3:MLKL shown in the box in panel **a**. Interface residues are shown as sticks; hydrophobic residues are shown as transparent surfaces; an electrostatic interaction is shown as a dashed line.

Fig. 5c, d). In contrast, a subset of mutations within the C-lobe interface in both RIPK3 and MLKL abrogated their capacity to reconstitute the necroptosis pathway in *RIPK3*^−/− or *MLKL*^−/− cells, respectively (Fig. 5b–f). While mutation of K331, R333, S373, S378, Q388, N416, and R421 of MLKL did not compromise their signaling capabilities, the V385W and S417D mutations prevented necroptosis from occurring, like the previously

described activation loop mutant of MLKL, T357E/S358E[51] (Fig. 5b). Because F386A and T357E/S358E human MLKL mutants behaved equivalently in previous studies, where necroptotic signaling was completely abrogated by either mutant[11,40,51], we included only one mutant—T357E/S358E MLKL—in this study as a control. RIPK3 mutants that abolish either the hydrophobic pocket into which MLKL V385 and F386

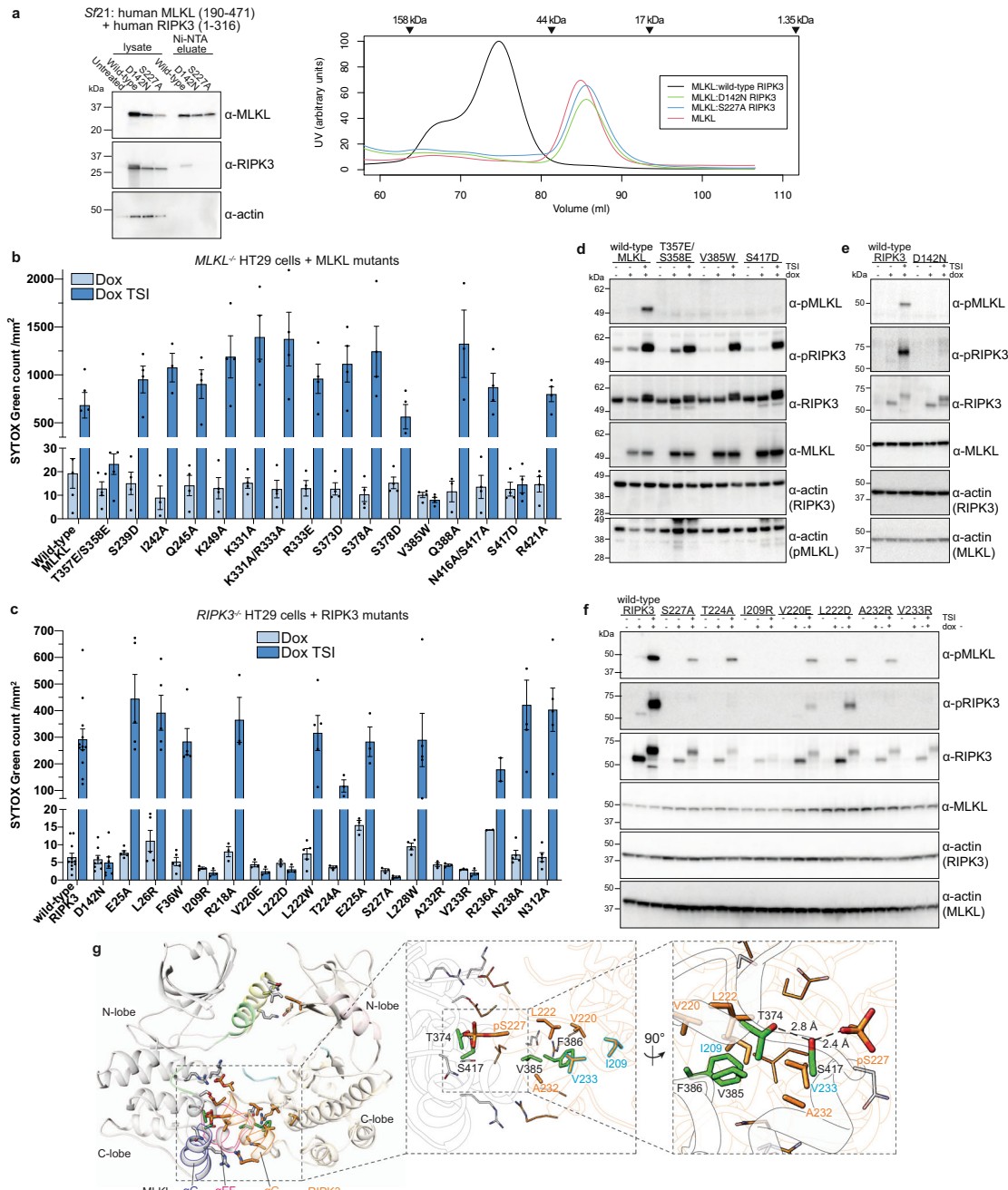

**Fig. 5 Bidentate C-lobe human MLKL:RIPK3 interactions govern necroptotic cell death. a** Human MLKL pseudokinase domain (190–471) formed a stable complex with wild type, but not D142N and S227A, human RIPK3 (1–316) in *Sf*21 cells. MLKL and RIPK3 were present in the lysates for all three constructs, but only wild-type RIPK3 coeluted with MLKL from Ni-NTA resin (left) or size exclusion chromatography (right). Elution volumes of molecular weight standards are shown above chromatograms. **b** V385W, S317D, and T357E/S358E MLKL failed to restore death signaling in *MLKL⁻/⁻* HT29 cells upon treatment with the necroptotic stimulus, TSI (T, TNF; S, Smac mimetic Compound A; I, pan-caspase inhibitor IDN-6556) for 20 h, as quantified by SYTOX Green uptake using IncuCyte imaging. **c** I209R, V220E, L222D, S227A, A232R, V233R, and the kinase-dead D142N mutant human RIPK3 were unable to restore death signaling in *RIPK3⁻/⁻* HT29 cells upon treatment with the necroptotic stimulus, TSI, for 20 h. **d** Wild-type MLKL, but not T357E/S358E, V385W, and S417D mutants underwent phosphorylation upon 20 h TSI stimulation when expressed in *MLKL⁻/⁻* HT29 cells. Wild type, but not kinase-dead D142N (**e**), and wild-type, S227A, T224A, V220E, L222D, and A232R, but not I209R and V233R (**f**), RIPK3 phosphorylated MLKL upon 20 h TSI stimulation when expressed in *RIPK3⁻/⁻* HT29 cells. **g** Crucial residues for human MLKL activation by RIPK3 are shown as sticks (MLKL, gray; αC helix, yellow; activation loop, green). Zoomed panels show crucial residues in the bidentate MLKL:RIPK3 interaction centered on MLKL F386 (hydrophobic) and RIPK3 pS227 (electrostatic; dashed lines). The RIPK3 residues, I209 and V233, whose substitution with Arg blocked MLKL phosphorylation and necroptosis signaling, are shown in cyan. Death data are presented as mean ± SEM, with four independent repeats for MLKL constructs, except *n* = 5 for wild type, T357E/S358E and *n* = 3 for I242A, Q388A MLKL; and three independent repeats for RIPK3 constructs, except *n* = 12 for wild type, *n* = 8 for D142N, *n* = 5 for E25A, L26R, F36W, L222W, L228W, N238A, N312A, and *n* = 2 for R236A RIPK3. Blots are representative of three independent experiments.

insert (I209R, V220E, L222D, A232R, V233R) or the residue interacting with MLKL S417, pS227 (via D142N or S227A mutation) failed to reconstitute necroptotic signaling when introduced into RIPK3$^{-/-}$ HT29 cells (Fig. 5c, e, f). We note the expression level of the I209R mutant was lower than that of wild-type human RIPK3 (Fig. 5f), which may reflect protein instability arising from the introduction of positive charge within a hydrophobic surface. Alanine substitutions of polar residues peripheral to this interaction (R218A, E225A, R236A, N238A, and N312A) and the second phosphorylation site observed in our complex structure (T224A), along with bulky substitutions of adjacent hydrophobic residues (L222W and L228W), in RIPK3 did not compromise reconstitution of necroptotic signaling. This indicates that, individually, none of these residues is essential for interaction with MLKL (Fig. 5c and Supp. Fig. 5c, d). Because most point mutants were capable of reconstituting necroptotic signaling, we assumed they would exhibit normal RIPK3:MLKL interactions and phosphorylation. Accordingly, we focused our further analyses of MLKL phosphorylation on interface mutants with defective necroptotic signaling. Intriguingly, while the V385W and S417D MLKL mutants both exhibited deficits in the hallmark of necroptotic pathway activation—MLKL S358 phosphorylation—only RIPK3 mutants I209R, V233R, and D142N abolished MLKL phosphorylation. Other RIPK3 mutants—V220E, L222D, S227A, and A232R—were able to phosphorylate MLKL despite failing to reconstitute necroptosis (Fig. 5c, f). I209 and V233 reside at the very interior of the hydrophobic pocket of RIPK3 (Fig. 5g), and introducing bulky polar sidechains by Arg substitution at these sites likely occludes MLKL F386 or V385 insertion into RIPK3. While the expression of the I209R mutant is lower than that of wild-type human RIPK3, which may impact its signaling capacity, collectively mutations to this pocket phenocopy the V385W (Fig. 5b) and F386A[11] MLKL mutations, which reciprocally prevent the interaction of MLKL with RIPK3. Introducing charges to the peripheral sites by V220E, L222D, or A232R substitution may attenuate, but not completely abrogate, V385 and F386 insertion, allowing transient interaction with and phosphorylation of MLKL. Similarly, the phospho-ablating S227A RIPK3 mutant also phosphorylated MLKL, albeit to a lesser extent, indicating that pS227 is not essential for phosphorylation of MLKL. Instead, these data and previous studies[4,25,47] argue for pS227 on RIPK3 serving an important role in mediating stable RIPK3:MLKL complex formation, which is necessary for recruitment of MLKL to the necrosome and engagement of auxiliary signaling proteins, including chaperones and the trafficking machinery[32,56,57]. Together, these data underscore the importance of human MLKL and RIPK3 C-lobe interactions in necroptotic signaling, where MLKL V385 and F386 serve as the crucial anchor that inserts into a hydrophobic pocket within RIPK3, augmented by additional polar contacts centered around RIPK3 pS227.

## Discussion

The interaction between the terminal effectors of the necroptosis cell death pathway, RIPK3 kinase and the MLKL pseudokinase, and the activation of MLKL's executioner function following RIPK3-mediated phosphorylation, are crucial checkpoints in the necroptosis pathway. Knowledge of precisely how human RIPK3 engages human MLKL has been restricted to deductions based on structures of mouse RIPK3 and the mouse MLKL pseudokinase domain:RIPK3 kinase domain complex. However, recent findings have revealed strict species specificity, such that mouse and human MLKL do not restore necroptotic signaling when introduced into MLKL-deficient human and mouse cells, respectively[39,40]. Here, we took a structural approach to deduce

the basis for species specificity by determining crystal structures of the human RIPK3 kinase domain alone or in complex with the human MLKL pseudokinase domain. These structures reveal that, as recently reported for human MLKL[47], human RIPK3 is conformationally plastic. Human RIPK3 bound to the Type I inhibitor, GSK′843, exhibits a classical active kinase conformation. In contrast, when in complex with MLKL, human RIPK3 adopts an open, inactive conformation. Because MLKL too is maintained in an open conformation, we propose this complex represents the inert complex of RIPK3 and MLKL that resides in the cytoplasm of cells prior to exposure to necroptotic stimuli.

The idea that human MLKL pseudokinase domain:RIPK3 kinase domain complex structure reported herein represents the basal, inactive pre-necroptotic complex is supported by studies of the complex using recombinant proteins and in cells. Previously, we reported a human MLKL pseudokinase domain binding monobody, Mb27, which detects endogenous, free MLKL in the cytosol only following necroptotic stimulation, because Mb27 binds an interface that overlaps with the RIPK3-binding site on MLKL[47]. Since Mb27 binding to MLKL was only detected post-necroptotic stimulation, we concluded that all MLKL was stably bound by RIPK3 under basal conditions, which occluded Mb27 binding to MLKL. Indeed, using another monobody, Mb32, MLKL was co-immunoprecipitated with RIPK3 under basal conditions, and RIPK3 binding was diminished post-necroptotic stimulation[47]. Unexpectedly, herein our biochemical and structural data indicate that phosphorylation of RIPK3 S227 is crucial for formation of this basal RIPK3:MLKL complex, and thus a precursor, rather than a barometer, of necroptotic activation. Intriguingly, within the human RIPK3:MLKL complex structure, RIPK3 S227 but not the MLKL activation loop T357/S358 was phosphorylated, consistent with the idea that RIPK3-mediated phosphorylation of the MLKL activation loop is a trigger for MLKL release from necrosomes in cells. Because the MLKL activation loop is distal to the RIPK3 activation loop in the human RIPK3:MLKL complex and, while in a distinct position and mostly unstructured, also remote in the mouse complex structure[46], we predict that a large change in conformation would be required for MLKL phosphorylation within a binary RIPK3:MLKL complex. It is possible that in the context of the necrosome, where RIPK3:MLKL subcomplexes are recruited and assemble into clusters, neighboring RIPK3 molecules could efficiently phosphorylate MLKL.

In both of the structures we report herein, tool compounds were required to aid crystallization, as apo human RIPK3 is highly unstable and does not crystallize readily. As expected, the Type I inhibitor, GSK′843, stabilized the active conformation of human RIPK3, and this structure can be used as a reference point for examining the conformation of human RIPK3 in the RIPK3:MLKL complex. Compound 10, however, a Type II inhibitor, bound in an unexpected mode. A compound from the same chemical series, with an identical central ring and tail group, Compound 9, has previously been crystallized with both mouse RIPK3 (PDB: 6OKO)[48] and human c-Met kinase (PDB: 3CE3)[58]. In both of these structures, the 4-fluorophenyl moiety inserts into the back pocket of the active site and flips the DFG motif out, in a classical Type II mode, unlike human RIPK3 within the complex structure where the 4-fluorophenyl moiety does not insert into the back pocket. We speculate that it is not Compound 10 that displaces the αC helix in the human RIPK3:MLKL structure but rather the compound accesses an unexpected binding mode by exploiting a cavity formed by the repositioning of the αC helix in the complex. Indeed, the mouse RIPK3 αC helix is positioned very similarly in the mouse MLKL:RIPK3 co-crystal structure (PDB: 4M69 (ref. [46]); Fig. 1c), where ATP rather than an inhibitory compound is bound.

In human MLKL, the activation loop appears to play a critical role in dictating the propensity for human RIPK3 binding. Mutation of the RIPK3 substrate residues in the human MLKL activation loop, T357/S358, to either alanine or glutamic acid, abrogated necroptotic signaling, presumably due to perturbed RIPK3 recognition[51]. Based on the complex structure reported herein, we propose that perturbation of the MLKL activation loop would impact the position of the MLKL αC helix, which would be expected to disrupt the N-lobe interaction with RIPK3. Furthermore, phosphorylation was recently proposed to promote conversion of human MLKL's pseudokinase domain from an open conformer, with the αC helix displaced by the activation loop helix akin to the conformation observed in our human RIPK3:MLKL complex structure, to a closed conformation with an intact K230–E250 salt bridge and an unstructured activation loop[47]. It is notable that we did not observe phosphorylation of the MLKL activation loop within our human RIPK3:MLKL complex structure, in keeping with the idea that phosphorylation of the human MLKL activation loop promotes structural interconversion and dissociation from RIPK3. Although no phosphorylated human MLKL structures have been reported to date, the reported crystal structure of the T357E/S358E phosphomimetic human MLKL pseudokinase domain (PDB: 6BWK)[51] provides a useful point of comparison. T357E/S358E human MLKL adopts a closed conformation, consistent with the idea that phosphorylation of human MLKL promotes conversion from the open, inactive form to the closed pro-necroptotic form.

Importantly, our structures of human RIPK3 bound to GSK′843 and the human RIPK3:MLKL complex provide a platform for rationalizing why human RIPK3 is unable to recognize and activate mouse MLKL (and vice versa). By mapping the residues equivalent to the human RIPK3:MLKL interface onto the mouse complex structure, we observed substantial divergence in the interacting residues. Only 12 of 29 MLKL and 20 of 34 RIPK3 residues are conserved between human and mouse complexes, while each protein exhibits around 60% sequence identity overall (Supp. Fig. 2). Besides differences in residues presented at each interface, the N-lobe interface is structurally most divergent between mouse and human complexes (RMSD = 0.987 Å over 110 Cα atoms across both chains). In both human and mouse MLKL complexed with RIPK3, the αC helix is the focal point of the N-lobe interaction interface, although the nature of the interaction differs markedly between mouse and human complexes. In the case of human MLKL, the αC helix protrudes owing to the adjacent, antiparallel activation loop helix, while in the mouse counterpart the αC helix is recessed and packs against the C-lobe orthogonal to the activation loop helix, which itself is positioned nearer to the C-lobe. In addition, the mouse MLKL αC helix is extended relative to that observed in the apo form, bends 40.5° around D238 where the interface ends, and presents F234 for π-π interactions with F27 in RIPK3's αC helix at the center of a largely hydrophobic interface. The latter interaction has been validated as important to mouse MLKL:RIPK3 binding[46], and the absence of these Phe residues in human RIPK3 and MLKL likely partly explains their incapacity to interact with their mouse cognates.

Previous studies have established the importance of the conserved MLKL C-lobe Phe/Tyr in RIPK3 engagement, in human (F386)[51], mouse (F373)[40,46], horse (Y385)[40], and viral (F202)[11] MLKL orthologs in binding to their cognate RIPK3 proteins. Our findings in the present study underscore the importance of the hydrophobic core of the C-lobe interface, because necroptotic cell death can be completely abrogated not only by the removal of the hydrophobic phenyl sidechain by F386A mutation[11] and Trp substitution of the adjacent residue V385 of human MLKL, but also by charged amino acid substitutions of five RIPK3

hydrophobic residues surrounding them. Because V385W mutation alters the shape of the interface without ablating hydrophobicity, the observation that this mutation is not tolerated suggests the precise shape of this hydrophobic core involving V385-F386 is essential. Because V385 is not conserved in mouse MLKL (substituted with P372), this may also contribute to the species-specificity among RIPK3:MLKL cognates.

The structure of the human RIPK3:MLKL complex indicates that C-lobe recognition relies on a bidentate interaction mode where one cluster of interactions centers on the aforementioned hydrophobic cleft, and another on interactions between human RIPK3 pS227 and human MLKL T374 (ref. [47]) and S417. Because acidic substitutions of each of these MLKL residues compromised necroptotic signaling, this raises the interesting prospect that their phosphorylation could serve as an additional regulatory mechanism to thwart necroptosis. To date, human MLKL T374 phosphorylation has been reported to occur in a cell cycle dependent manner[59,60], although the kinase responsible for this modification, and whether MLKL S417 undergoes phosphorylation in cells, remain to be determined[25]. Previous biochemical studies implicated the RIPK3 αG helix as a crucial determinant of MLKL species specificity, where chimeras of mouse RIPK3 harboring the human RIPK3 αG helix were able to immunoprecipitate human MLKL[61], although whether signaling could be restored was not examined. Interestingly, both mouse and human RIPK3 have an additional pThr within their αG helices, although their positions are conserved neither in sequence nor structure between species. While human RIPK3 pT224 interacts electrostatically with R333 of human MLKL, mouse RIPK3 pT231 forms no apparent interactions in the co-crystal structure. Consistent with the heterogeneous locations, our data here and those of others[4,46], indicate that pT224 in human RIPK3 and pT232 in mouse RIPK3 are not essential to MLKL recognition and activation, and instead likely serve an auxiliary function. However, along with several other electrostatic interactions peculiar to the interface between human RIPK3 and MLKL C-lobes, the human MLKL R333:RIPK3 pT224 interaction would be expected to contribute to species selectivity. Strikingly, none of the salt bridges between the C-lobes of human RIPK3 and MLKL are conserved in the mouse RIPK3:MLKL complex structure (Fig. 1d, e). These include the human MLKL K372:RIPK3 E226, MLKL K372:RIPK3 pS227, and MLKL E418:RIPK3 R236 salt bridges, which combined with the MLKL K333:RIPK3 pT224 interaction, present a distinct electrostatic landscape in the human RIPK3:MLKL C-lobes, which may explain the species specificity of RIPK3:MLKL recognition.

Collectively, these data indicate that the human RIPK3:MLKL complex structure reported herein is representative of the basal, inactive form of both RIPK3 and MLKL that occurs in the cytoplasm of cells prior to exposure to necroptotic cues (summarized in Fig. 6). In this complex, RIPK3 and MLKL both adopt open, inactive conformations. We propose that following induction of necroptosis, this complex is recruited to the necrosome where activated RIPK3 can phosphorylate the activation loop of MLKL to promote MLKL's interconversion to a closed, pro-necroptotic form, which can disengage from the necrosome and assemble into killer oligomers. MLKL recruitment to the necrosome appears to be an essential prelude to liaison with chaperones and the trafficking machinery[31,32,56,57], which facilitate translocation to the plasma membrane where MLKL's 4HB executioner domain can permeabilize membranes to inflict cell death. Our mutational studies indicate that disruption of the stable assembly of basal complex, such as by abolishing phosphorylation of S227 in RIPK3, is sufficient to completely abrogate necroptotic cell death. While disruptive mutants retain the capacity to transiently engage and phosphorylate the human

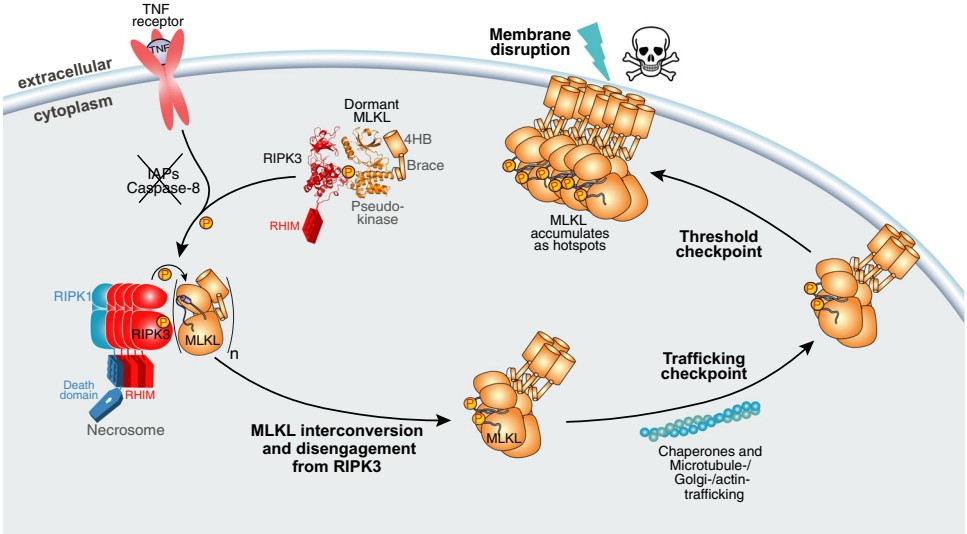

**Fig. 6 A model of the transition of human MLKL from a dormant form to necroptotic effector.** Human MLKL resides in a dormant state stably-complexed to RIPK3 in the cytoplasm via the bidentate interaction between MLKL and RIPK3 C-lobes. Following receipt of a necroptotic stimulus, RIPK3:MLKL complexes are recruited to the necrosome where RIPK3-mediated phosphorylation promotes MLKL structural interconversion, dissociation from the necrosome, and oligomerization[47]. Mutational perturbation of the MLKL:RIPK3 interaction prevents their stable association and recruitment to the necrosome, identifying formation of stable MLKL:RIPK3 complexes as a crucial checkpoint in the necroptotic pathway. Recruitment to the necrosome allows engagement of chaperone- and Golgi/microtubule/actin-mediated pathways to facilitate MLKL translocation to the plasma membrane[32,56,57]. Upon reaching a threshold level at the plasma membrane, MLKL can permeabilize membranes to cause cell death and damage-associated molecular pattern (DAMP) release. The skull and crossbones image (Mycomorphbox_Deadly.png; by Sven Manguard) was used under a Creative Commons Attribution-Share Alike 4.0 license.

MLKL activation loop, our data indicate that such a transient interaction, and phosphorylation of MLKL, are insufficient to facilitate necroptotic cell death. These studies highlight the assembly of the basal RIPK3:MLKL complex and its recruitment to the necrosome as key checkpoints in the necroptosis pathway, and raise the prospect that pharmacological targeting of these crucial signaling events could counter diseases arising from errant necroptosis, such as inflammatory diseases of the lung, gut and kidney.

## Methods

**Expression constructs.** Mutations were introduced into a human MLKL DNA template (from DNA2.0, CA) using oligonucleotide-directed overlap PCR or were synthesized by ATUM (CA) and subcloned into pF TRE3G PGK puro[6] as *Bam*HI–*Eco*RI fragments. Genes encoding N-terminally FLAG-tagged wild type or mutant human RIPK3 were either generated using oligonucleotide-directed overlap PCR from a human RIPK3 DNA template (ATUM) or synthesized by ATUM and ligated into pF TRE3G PGK puro. Oligonucleotide sequences are presented in Supplementary Table 2. Insert sequences were verified by Sanger sequencing (Micromon DNA Sequencing Facility, VIC, Australia). Vector DNA was co-transfected into HEK293T cells with pVSVg and pCMV ΔR8.2 helper plasmids to generate lentiviral particles, which were transduced into *MLKL*[−/−] [51] or *RIPK3*[−/−][47] HT29 cells and selected for genomic integration using puromycin (1.25 μg/mL for *MLKL*[−/−] cells, 2.5 μg/mL for *RIPK3*[−/−] cells; StemCell Technologies) using established procedures[6,28,51].

**Reagents and antibodies.** Primary antibodies used in this study were: rat anti-MLKL (clone 3H1, produced in-house; 1:1000 dilution; available as MABC604, EMD Millipore, Billerica, MA, USA), rat anti-human MLKL pseudokinase domain (clone 7G2, produced in-house[32]; 1:2000 dilution; soon available from EMD Millipore as MABC1636), rabbit anti-human MLKL phospho-S358 (AB187091, Abcam; 1:2000), mouse anti-Actin (C4) HRP (sc-47778 HRP, Santa Cruz Biotechnology; 1:10000), rat anti-human RIPK3 (clone 1H2, produced in-house[11]; 1:1000; available as MABC1640, EMD Millipore, Billerica, MA, USA), rabbit anti-human RIPK3 phospho-S227 (D6W2T, CST; 1:2000). Recombinant hTNF-Fc[62] was produced in-house, while the Smac mimetic, Compound A[63] and the pan-caspase inhibitor, IDN-6556/Emricasan, were provided by Tetralogic Pharmaceuticals. The RIPK3 inhibitors, GSK′843 (ref. [54]) and BMS Compound 10 (ref. [48]), were kindly provided by Anaxis Pty Ltd (Australia).

**Recombinant protein expression and purification.** Human MLKL pseudokinase domain (residues 190–471): human RIPK3 kinase domain (residues 1–316; harboring C3S and C110A substitutions to prevent disulfide bond formation) complex was co-expressed and purified from *Sf*21 insect cells using an established procedure[47]. The co-expression bacmid was prepared in DH10MultiBac *Escherichia coli* (ATG Biosynthetics) from a pFastBac Htb vector encoding a TEV protease-cleavable His$_6$ fusion N-terminal to the human MLKL pseudo-kinase domain (residues 190–471), with a p10 promoter-human RIPK3 (residues 1–316; C3S, C110A)-HSV TK poly(A) signal (from pAceBac2; ATG Biosynthetics) cassette introduced into pFastBac Htb:human MLKL (190–471) by In-Fusion (Takara Bio) cloning into the non-coding region between BbsI and SnaBI sites of the pFastBac Htb vector backbone. A bacmid encoding human RIPK3 (2–316; C3S, C110A) with an N-terminal, TEV protease-cleavable His$_6$ tag was prepared from pFastBac Htb via recombination in DH10MultiBac *E. coli* cells. *Sf*21 insect cells were cultured in Insect-XPRESS (Lonza) media. Bacmids were introduced into *Sf*21 cells by Cellfectin II (Thermo Fisher Scientific) mediated transfection in six-well plates using the Bac-to-Bac protocol (Thermo Fisher Scientific) as detailed previously[64]. The resulting P1 baculovirus was harvested after 4 days static incubation at 27 °C and added at 1% v/v to 100 mL *Sf*21 cells at $1.5 \times 10^6$ density, which were shaken at 27 °C, 130 r.p.m. for 4 days before the supernatant P2 virus was harvested. 50 mL P2 virus was added to 0.5 L *Sf*21 ($2.5–3 \times 10^6$ cells/mL) in 2.8 L Fernbach flasks, and cells cultured for 48 h at 27 °C, 90 r.p.m., using an established procedure[65]. Cells were harvested at $500 \times g$ and pellets snap frozen in liquid N$_2$ and either thawed immediately for lysis or stored at −80 °C until required. Cells were lysed by sonication in 0.5 M NaCl, 20 mM Tris-HCl, 20% v/v glycerol, 10 mM imidazole (pH 8), 0.5 mM TCEP [tris-(2-carboxyethyl)phosphine] supplemented with EDTA-free cOmplete Protease Inhibitor (Roche). Throughout purification, the proteins were maintained at 4 °C. The lysate was clarified by centrifugation ($40,000 \times g$, 30 min) and then mixed with HisTag Ni-NTA resin (Roche), which has been pre-equilibrated with low imidazole buffer [0.5 M NaCl, 20 mM Tris-HCl (pH 8.0), 5 mM imidazole (pH 8.0), 20% v/v glycerol] for 1 h at 4 °C before the beads were pelleted at $500 \times g$ and washed extensively with low imidazole buffer and a wash buffer containing 35 mM imidazole. The protein was eluted in high imidazole buffer [0.5 M NaCl, 20 mM Tris-HCl (pH 8.0), 250 mM imidazole (pH 8.0), 20% v/v glycerol]. The eluted His$_6$-RIPK3 protein or His$_6$-MLKL:RIPK3 complex were further purified by cleaving the His$_6$ tag with TEV protease, dialysis, Ni-pulldown to eliminate uncut material and the TEV protease, and application to a HiLoad 16/160 Superdex 200 column (Cytiva) pre-equilibrated with SEC buffer [20 mM HEPES (pH 7.5), 200 mM NaCl, 5% v/v glycerol]. Purified fractions, as assessed following resolution by reducing StainFree SDS-PAGE gel electrophoresis (Bio-Rad), were pooled and concentrated to 5 mg/mL using a 30 kDa MWCO spin concentrator (Millipore), aliquoted, and snap frozen in liquid N$_2$ for storage at −80 °C.

**Protein crystallization and structure determination**. MLKL:RIPK3 complex was incubated overnight at 4 °C with a fivefold molar excess of BMS Compound 10, and RIPK3 was incubated with fivefold excess of GSK′843 before excess compound was eliminated using 30 kDa MWCO spin concentrators. Proteins were subjected to sitting drop crystallization trials conducted at the C3 Facility (Parkville, VIC) with 150 nL protein solution mixed with 150 nL reservoir volume in sitting drops, with 50 μL reservoir volume. RIPK3 kinase:MLKL pseudokinase domain crystals grown in 0.2 M ammonium acetate, 30% w/v polyethylene glycol 4000, 0.1 M trisodium citrate-citric acid buffer (pH 5.5) at 8 °C, and RIPK3 kinase domain crystals grown in 0.2 M magnesium chloride, 25% w/v polyethylene glycol 3350, 0.1 M Tris (pH 8.5) at 8 °C were soaked in cryoprotectant (reservoir solution supplemented with 20% v/v ethylene glycol) and flash-frozen in liquid nitrogen. MLKL:RIPK3 complex X-ray diffraction data were collected at the Australian Synchrotron MX2 beamline using the EIGER X 16M detector[66] with in-house MX control software. The MLKL:RIPK3 complex diffraction data were indexed and integrated in XDS[67], and merged and scaled in AIMLESS[68]. After adding $R_{free}$ reflections using phenix.reflection_file_editor[69], phases were solved for the MLKL:RIPK3 complex by molecular replacement in PHASER[70]. To minimize model bias, molecular replacement with structural models in all available conformations were attempted, including the closed (PDB: 4MWI) and open (PDB: 7JXU) human MLKL conformers[47,50], and the closed (PDB: 4M66) and open (PDB: 4M69) mouse RIPK3 conformers[46]. The solution using the human MLKL pseudokinase domain structure from the Mb32 complex (PDB: 7JXU)[47] and the mouse RIPK3 kinase domain from the mouse RIPK3:MLKL complex structure (PDB: 4M69)[46] were selected based on reporting the highest LLG score and the closest match between the molecular replacement model and the electron density map. Human RIPK3:GSK′843 diffraction data were processed using DIALS[71] and autoPROC[72], and elliptically truncated using STARANISO owing to severe anisotropy, with phases solved by molecular replacement. The default I/sigI threshold of 1.2 was used to define the resolution for elliptical truncation, where the cutoff for each axis was 5.7, 3.7, and 3.1 Å in the a*, b*, and c* axes, respectively. The highest resolution reflection present within this cutoff was at 3.23 Å. $R_{free}$ flags were added using phenix.refine post molecular replacement. COOT and phenix.refine were used for iterative real-space and reciprocal-space refinement by manual model building and phase refinement[73,74]. Protein interfaces and the biological assembly in solution were analyzed using PDBePISA[55]. Interface area is defined as the difference in total accessible surface areas of isolated and interfacing structures divided by two in Å². Structure models were validated using MolProbity[75]. Structural figures were generated using either UCSF Chimera[76] or PyMOL (The PyMOL Molecular Graphics System, Schrödinger, LLC.). Structure alignments were calculated using UCSF Chimera. Data collection and processing statistics are summarized in Table 1.

**Small angle X-ray scattering**. SAXS data were collected using the inline-SEC setup at the Australian Synchrotron SAXS/WAXS beamline. A co-flow setup was implemented to maximize signal-to-noise by allowing exposure of protein to high flux X-rays without direct contact with the capillary wall, while minimizing sample dilution[77,78]. RIPK3:MLKL complex (unliganded or pre-incubated with Compound 10) was eluted from a Superdex 200 5/150 Increase column (Cytiva) in 200 mM NaCl, 20 mM HEPES pH 7.5, 5% glycerol at a flow rate of 0.2 mL/min, where the SEC eluate passes through a capillary in the path of the SAXS beam and scattering data were collected in 1 s exposures serially over the course of the chromatography run. Data were reduced, 2D radially integrated, and background scatter subtracted using ScatterBrain v2.82 (developed in-house by Stephen Mudie). The background scatter was calculated by averaging exposures from the SEC run prior to protein elution, and was subtracted from the scatter arising from the protein peak off the SEC column. Exposures from the apex of the protein elution peak were averaged and subjected to analyses using the ATSAS 3.0 package[79] including Guinier analyses in PRIMUS, $P_r$ in GNOM, and comparisons with theoretical and experimental scatter using CRYSOL. Data collection and processing statistics are summarized in Supplementary Table 1.

**Cell culture**. The human colorectal adenocarcinoma HT29 (a gift from Mark Hampton), and their $MLKL^{−/−}$[51] and $RIPK3^{−/−}$[47], counterparts were cultured in human DMEM (Thermo Fisher) media supplemented with 8% vol/vol fetal calf serum (FCS; Sigma), with puromycin (2.5 μg/mL; StemCell Technologies) added for lines stably transduced with inducible MLKL or RIPK3 constructs. Routine PCR testing confirmed cell lines to be mycoplasma-negative.

**IncuCyte cell death assay**. HT29 cells were seeded into 48-well plates at $3 \times 10^4$ cells/well and left to settle for 16–24 h prior to treatment with doxycycline (20 ng/mL) overnight to induce expression of the relevant MLKL construct. RIPK3 expression was induced comparably but with 2.5 ng/mL doxycycline. Cells were then treated with TNF (100 ng/mL), the Smac-mimetic compound A (500 nM) and the pan-caspase inhibitor IDN-6556 (5 μM) to induce necroptosis in FluoroBrite DMEM (Thermo Fisher Scientific) media supplemented with 1% FCS (Sigma), 1 mM Na pyruvate (Thermo Fisher Scientific), 1 mM L-GlutaMAX and SYTOX Green (Thermo Fisher Scientific) using an established protocol[32,47]. Cells were imaged using default bright field and green channel settings on ×10 objective in an

IncuCyte S3 System (Essen Bioscience) with scans every hour for 24 h. The number of SYTOX Green-positive cells per mm² over time was quantified using IncuCyte S3 v2020C Rev 1 software (Essen Bioscience).

**Western blots**. HT29 cells were seeded into 24-well plates at $7 \times 10^4$ cells/well and induced overnight with 20 ng/mL doxycycline for MLKL expression, or 2.5 ng/mL doxycycline for RIPK3 expression. Cells were then treated with TNF (100 ng/mL), the Smac-mimetic compound A (500 nM) and the pan-caspase inhibitor IDN-6556 (5 μM) to induce necroptosis in DMEM media supplemented with 1% FCS. MLKL expressing cells were harvested after 3 h and RIPK3 expressing cells were harvested 7.5 h post-necroptotic induction (with TSI) in 2× SDS Laemmli lysis buffer, boiled at 100 °C for 5–10 min, and then resolved by 4–15% Tris-Glycine gel (Bio-Rad) or 4–12% Bis Tris (Invitrogen) gels. Proteins were transferred to PVDF membrane, blocked with 5% w/v skim milk powder in TBST, and probed with antibodies as indicated.

**Thermal shift assays**. Thermal shift assays were performed using a Corbett Real Time PCR machine with proteins diluted in SEC buffer to 10 μM and assayed with either 40 μM Compound 10 or an equivalent volume of DMSO control in a total reaction volume of 25 μL using established methods[65,80]. SYPRO Orange (Sigma-Aldrich) was used as a probe with fluorescence detected at 530 nm. The temperature was raised in 1 °C per min steps from 25 °C to 95 °C and fluorescence readings were taken at each interval. For each well, sample fluorescence was plotted as a function of increasing temperature. The melting temperature ($T_m$) corresponding to the midpoint for the protein unfolding transition was calculated by fitting the sigmoidal melt curve to the Boltzmann equation using non-linear least square fit with R Studio, with $R^2$ values of >0.99. Data points after the fluorescence intensity maximum were excluded from fitting. Three technical replicates were performed and plotted for each protein and ligand, and the mean $T_m$ are shown for each.

**Reporting summary**. Further information on research design is available in the Nature Research Reporting Summary linked to this article.

## Data availability

Source data are provided for western blots and cell death assays in Fig. 5 and Supp. Fig. 5 and thermal stability shift assays in Supp. Fig. 1a. All data, including expression construct sequences, are available from the corresponding authors upon request. The atomic coordinates for the human RIPK3 kinase domain in complex with GSK′843, and in complex with the human MLKL pseudokinase domain and Compound 10, have been deposited in the Protein Data Bank with the accession numbers, 7MX3 and 7MON, respectively. The atomic coordinates for previously reported structures used in this study are available from the Protein Data Bank with accession codes: 4MWI, 7JXU, 7JW7, 4M66, 4M69, 6OKO, 3CE3, 6BWK. Source data are provided with this paper.

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

## Acknowledgements

We thank Dr. Janet Newman and the C3 Facility for assistance with screening for protein crystallization conditions, and the Australian Synchrotron MX and SAXS/WAXS beamline staff for assistance with data collection. This research was undertaken using the MX2 crystallography beamline at the Australian Synchrotron, Victoria, Australia, and made use of the ACRF Detector. We acknowledge scholarship support for Y.M. (Melbourne Research Scholarship; AINSE PGRA scholarship) and S.E.G. (Australian Government Research Training Program Stipend Scholarship; Wendy Dowsett Scholarship). We are grateful to the National Health and Medical Research Council for fellowship (G.L., 1117089; P.E.C., 1079700; J.M.M., 1172929), grant (A.L.S and J.J.S., 2002965) and infrastructure (IRIISS 9000653) support; Anaxis Pharma Pty Ltd for funding support; the Australian Cancer Research Foundation; and the Victorian Government Operational Infrastructure Support scheme.

## Author contributions

Y.M. and K.A.D. designed, performed, and analyzed experiments, performed structural studies and wrote the paper with J.M.M.; C.F., S.N.Y., C.R.H., C.L., L.-Y.L., and A.D.C. designed and performed experiments; S.E.G., A.L.S., J.-M.G., G.L., and J.J.S. analyzed and interpreted data; P.E.C. and J.M.M. designed the project, analyzed, and interpreted data; all authors commented on the manuscript.

## Competing interests

Y.M., K.A.D., C.F., S.N.Y., S.E.G., C.L., J.-M.G., L.-Y.L., A.D.C., A.L.S., G.L., P.E.C., and J.M.M. contribute, or have contributed, to a project developing necroptosis inhibitors in collaboration with Anaxis Pharma Pty Ltd. The remaining authors declare no competing interests.
