## [Peer Review File · Nature Communications]

Human RIPK3 maintains MLKL in an inactive conformation
prior to cell death by necroptosisREVIEWER COMMENTS

Reviewer #1 (Remarks to the Author):

Necroptosis, a regulated form of cell death, is executed by activating RIPK3 and MLKL through phosphorylation of RIPK3. However, the detailed molecular mechanisms of how RIPK3 activates MLKL are not fully understood yet. Moreover, There is an interspecies divergence of RIPK3 and MLKL; human RIPK3 activates human MLKL, whereas murine RIPK3 and murine MLKL. A previous study reported that crystal structures of murine RIPK3 and murine MLKL. In this manuscript, Meng et al. reports the crystal structure of the human RIPK3 kinase domain alone and in complex with the human MLKL pseudokinase domain. They find a distinct interface in the human RIPK3:MLKL complex, allowing the human RIP3/MLKL complex to remain inactive compared to murine one. Moreover, mutational analysis validated interface residues within the C-lobes of human RIPK3 and MLKL as crucial for necroptotic signaling in human cells.

The present study elucidates the difference between human and murine RIPK3/MLKL complexes and provides a cue to investigate the mechanisms of how the inactive RIPK3/MLKL complex is activated. The reviewer requests several additional experiments and discussions to improve the manuscript. The followings are specific comments.

Major points:

1. The authors propose a model that the RIPK3/MLKL-containing complex remains inactive in the cytosol before stimulation, and the complex becomes dissociated after stimulation. It is unclear whether free RIPK3 or free MLKL exists in the cells before or after stimulation. The authors need to show the stoichiometry of the RIPK3/MLKL complex, free RIPK3, and free MLKL in the cells. For example, Western blotting analysis using cell lysates before and after immuno-depletion of the RIPK3/MLKL complex with anti-RIPK3 or anti-MLKL antibodies may be feasible for evaluating the amounts of cellular pools of free RIPK3 or free MLKL.
2. Lines 104-111; The authors claim that human RIPK3 S227 phosphorylation is required for the interaction of MLKL. Assuming that S227 phosphorylation is mediated by RIPK3 itself, RIPK3 must be activated when generated in Sf9 cells. However, T357 and S358 of MLKL are not phosphorylated in the complex. The authors mention that "human MLKL T357 and S358, are distal from the catalytic HRD motif of RIPK3 (Supp. Fig. 3f), and a conformational change would be required for their phosphorylation to occur" (lines 259-261). In contrast, a previous study reported that both murine RIP3 and MLKL are phosphorylated in the complex (Cell Reports 5, 70-78, 2013). Considering the distance between the activation loop of murine MLKL and the catalytic HRD motif of murine RIPK3, the authors need to explain the difference in more detail.
3. Lines 150-162; The authors mention that K230 in the β 3-strand in the N-lobe is proximal to Q356 in the activation loop helix in the open conformer of human MLKL. The authors previously reported that a similar proximal interaction of K219 and Q343 is detected in murine MLKL, and mutations of these residues result in a constitutive active mutant of MLKL (Immunity 39, 443-453, 2013). Given that topology of the α C helix and the activation loop between murine and human MLKL is different, it would be interesting to test whether mutation of these amino acids (K230 and Q356) of human MLKL may affect MLKL' function.
3. In Figure 5c, d, e, the signals of pMLKL are too weak. The authors need to change them to more representative ones. Moreover, the authors should show pRIPK3 as well as total RIPK3.
4. In Figure 5e, it is unclear why RIPK3 S227A cells do not undergo necroptosis even though phosphorylation of MLKL is induced. The authors mention that "such a transient interaction and phosphorylation of MLKL are not sufficient to facilitate necroptotic cell death" (lines 415-419). If the authors claim "transient interaction and phosphorylation of MLKL", the authors should investigate the kinetics of TSI-induced phosphorylation of MLKL and quantify the signaling intensities of pMLKL in RIPK3 S227A cells compared to RIPK3 WT cells following TSI stimulation.

5. In Figure 5e, the expression of RIPK3 I209R is very low following dox treatment. If the expression levels of RIPK3 I209R are very low, it may be unfair to discuss whether this RIPK3 mutant is functional in terms of necroptosis induction. The authors need to discuss this point, alternatively change the Figures to more representative ones.

Minor point:

1. Line 254, P36 should be F36.
2. Line 297, there is no data of F386A mutant in Figure 5. The authors need to include it.
3. In Supplementary Figure 3, Figure 3e should be labeled Figure 3f. Figure 3e is missing.
4. In Supplementary Figure 4a, TSEE should be spelled as T357E/S358E.
5. In Supplementary Figure 4d, I109R should be changed to I209R.

Reviewer #2 (Remarks to the Author):

The manuscript presents the crystal structures of human RIPK3 kinase domain alone and in complex with the MLKL pseudokinase domain. These are critical protein players in necroptotic cell death. MLKL is the terminal effector in necroptosis signaling pathway, and is recruited to RIPK3 and subsequently phosphorylated by RIPK3. Importantly, there are sequence and functional variability in the RIPK3/MLKL pairs between species, which is why determination of the structures of the human proteins is an important step toward the mechanistic understanding how this pathway is regulated.

The study is presented well, and it is satisfying that the authors put the findings in context of previous work in the field so as to highlight its own importance. Without this, it may seem that the structural finding is a mere incremental improvement over the previously determined structure of the protein complex from mouse especially since the relative overall orientation of the two kinase domains is similar. However, determination of the human protein complex should provide a definitive molecular-level viewpoint for future mechanistic studies of this important pathway.

In general, the structural studies of the RIPK3:MLKL complex are satisfactory, with some specific questions raised below:

1. was autobuild attempted to model this structure? At 2.2 Å resolution, it is possible that autobuild is achievable, which should remove model bias toward the original molecular replacement models.
2. Related, how have the authors ensured that model bias has not been introduced with respect to the relative kinase domain lobe orientations (active versus inactive). Was molecular replacement or rigid body refinement conducted with the individual lobes?

The structure of the open conformation of RIPK3 alone (in complex with an inhibitor) is also presented. I have some reservations of this structure, namely the accompanying statistics and data treatment, in its current form. However, I hope that the authors will be able to address most of my questions directly in rebuttal and with a few additions/changes to the structure and/or manuscript text.

The authors state that the diffraction data for the RIPK3/GSK483 structure suffered from anisotropy, and so ellipsoidal truncation was performed automatically in autoPROC with STARANISO. This is an acceptable treatment of the data; however, a few details could be provided which would further help explain this anisotropy effect and ensure that the data processing is appropriate. I would look forward to re-examining this structure if the following points could be addressed.

Major questions:

1. How was the resolution limit reached? Did the authors carefully examine a F/σ vs resolution

plot for each of the three principal axes to reach the resolution limit? Related, what was the F/σ of the discarded reflections?

2. I have concerns with the low completeness of even the ellipsoidal truncated data (90.7%, 56.9%). Does the completeness improve if a slightly lower (poorer) resolution cutoff is chosen (i.e. 3.3 Å vs 3.2 Å)? Or can the authors provide rationale (with references, if applicable) that it is acceptable for anisotropic and truncated data to have such low completeness? In general, the quality of an electron density map can suffer greatly by low completeness. Is the same true for anisotropic data?

3. Related to resolution limit and completeness, have the authors examined electron density maps calculated at lower resolution limits? Can better maps be obtained by different treatment of the data? The maps as presented in Fig S4 are quite poor even for 3.2 Å resolution. The resolution limit should not be pushed unnecessarily especially if the map quality is better achieved at lower resolution cutoff.

4. Similar to questions about the RIPK3:MLKL complex structure, was molecular replacement and/or rigid body refinement conducted with the individual kinase domain lobes to ensure that their relative orientation (active vs inactive) has not suffered from model bias, especially here at lower resolution and with anisotropic data?

5. Are the R_{merge} and I/σ values in Table 1 for 7MX3 from after truncation? Related, can the authors provide R_{merge} , completeness, and I/σ values for both before and after anisotropic truncation?

6. The authors could run an orthogonal analysis of their original scaled reflection file to validate the choice of resolution cutoff. Namely, the UCLA-DOE LAB — Diffraction Anisotropy Server. This would help cross-validate the cutoff determined by STARANISO.

7. Can the authors provide details as to how/when during processing the R_{free} reflections were chosen?

8. The Clashscore and Ramachandran distribution could be improved.

9. Does the disulfide bond between the alpha-G helices of apposing copies induce a conformational change? This is important, since the manuscript states that “the most substantial structural changes upon MLKL complexation occur on and around the alphaG helix”, and because the disulfide forms between Cys residues on the alphaG helix.

Several additions to the SAXS data section should also be considered:

1. The authors should include one or more molecular weight determinations from the SAXS data, namely the Porod Volume, volume of correlation, etc. Is the molecular weight consistent with what is expected for the complex? This can be added to Table S1. These types of calculation are readily available in many SAXS software packages (e.g. RAW) and online servers (SAXSMoW).

2. Based on the “good fit” of the theoretical curve to the crystal structure, it seems that calculation of a bead model and/or electron density (DENSS) would make for a descriptive figure when superposed with the crystal structure. The authors might consider including such a model to Fig S1, though this is not absolutely required.

Additional general questions/comments on manuscript:

1. Intro paragraph 1, lines 33-37, can this sentence be better clarified?

2. Can the phosphorylation of RIPK3, and lack of phosphorylation of MLKL, verified in vitro?

3. Supp Fig 1c difference density map, what is the contour level shown?

4. Thermal shift assay: can another kinase as negative control be tested?

5. Was AMPPNP tested instead of inhibitor for complex formation?

6. Can the change between mouse and human (visualized in Fig 2c) be explained by the binding of compound 10? Does compound 10 bind mouse MLKL?

Overall, the manuscript is well written and should be understandable to a broad audience. The questions raised in this review will need to be addressed before the entirety of the structural data and interpretation are acceptable for publication. However, based on what is presented in its current form, I hope and expect that the authors will be able to address the issues and that the biological implications of the work, and therefore the message of the manuscript as a whole, will largely be upheld.

Reviewer #3 (Remarks to the Author):

In current work, the authors have determined the structure of human RIPK3 kinase domain alone and in complex with MLKL pseudokinase. Current complex structure was compared with mouse counterpart, the structure of mouse RIPK3-MLKL complex. The authors identified how human RIPK3 maintains MLKL in an inactive conformation prior to induction of necroptosis and residues involved in necroptotic signaling were identified and characterized using human cells.

Overall work has good quality. However, major revisions are needed before current manuscript be suitable for publication in current journal. I have following concerns in current manuscript

-English grammar should be checked throughout the manuscript. Many paragraphs are not written clearly and do not explain the meaning of sentences. In single sentence, many results and interpretations are written and merged together with no clarity and grammar.

-Human RIPK3 (residue 2-316; C3S, C110A)+GSK'843 complex structure was determined at low resolution (~3.23 Å) and compared with high resolution (~2.23 Å) structure of Human RIPK3 (residues 1-316; C3S, C110A)+human MLKL (residues 190-471) complex structure. For comparative structure analysis, it would be better if author should have high resolution structure of Human RIPK3-GSK'843 complex

Abstract section:

Page 2: lines 15-20 contains only background and rationale behind current study, while lines 20-25 contains very little information about results and significance of current work. The authors should outline clearly critical findings and their implications discovered from current work in the Abstract section.

Page 2, lines 16-17: How pathway involved in dysregulation of necroptosis in inflammatory diseases could offer as key therapeutic target?

Introduction section:

Page 4, lines 69-70: ...“RIPK3 and MLKL bound in a face-to-face arrangement with apposing activation loops”..., Please rewrite the current sentence and explain clearly the meaning of face to face arrangement.

Page 4, lines 76-77: Meaning of the paragraph starting with” The activation loop..... RIPK3 N-lobe binding” is not clear and should be rewritten.

Page 4, lines 78-79: Please include briefly here the critical finding obtained from mutational analysis of interface residues within the C-lobes of the mouse RIPK3-MLKL complex, if available.

Results section:

Page 5, lines 86-88: Why individually purified human RIPK3 and human MLKL did not assemble as complex, when expressed individually and mixing them together, as observed in co-expression?
-Did authors got some clues in comparative structure analysis between human RIPK3 with human RIPK3+MLKL complex structures

Page5, lines 92-94: What is the intrinsic instability in human RIPK3, which prevented the human RIPK3:MLKL complex crystallization?

-How compound-10 helped in binary complex formation?

Page 5, lines 106-108: What could be the possible role of T224 phosphorylation in RIPK3, as S227 phosphorylation is designated as determinant of MLKL interaction.

Page 6, lines 113-114: Compound-10 is only specific for RIPK3 ATPase pocket. How high concentration of compound-10 promote binding to MLKL ATPase pocket? Do both proteins have similar ATPase pockets?

Page 6, lines 137-138: Did authors observed any change in globular shape of SAXS envelope of human RIPK3:MLKL complex in presence and absence of compound-10 ?
-If not, is it due to fact they used higher concentration of Glycerol in SAXS buffer?

Page 7, line 167: Salt bridge and hydrophobic “regulatory” (R)-spine interactions are hallmarks of an inactive MLKL open conformation, when bound to RIPK3 and similar conformation was observed in Mb32 bound MLKL (PDB: 7JXU).

-Do the conformational changes in MLKL were identical in RIPK3 and Mb32 bound states?

Page 7-8, lines 173-176:

Did structural changes in MLKL upon Phosphorylation were identical to structural changes in MLKL upon complex formation with RIPK3? that lead RIPK3:MLKL complex dissociation?

Page 10, lines 260-261: pS227 and pT224 of RIPK3 form salt bridges with K372 and R333 of MLKL and appear integral to MLKL binding.

These salt bridges are the only key interactions in MLKL binding or other interactions also contribute in binding specificity?

Methods section:

Page 19, line 509-510: SAXS data collected using buffer (200 mM NaCl, 20 mM HEPES pH 7.5, 5% glycerol). 5% Glycerol is high for SAXS data, as it makes buffer matching difficult and decreases the contrast? and be repeated.

Figures legends section:

Page 35, lines 838-840, Fig. 1C legend: How human RIPK3 (grey):MLKL complex was superposed on mouse RIPK3:MLKL complex structure?

Page 37, lines 857-858, Fig 2b legend: How human MLKL (blue):Mb32 (teal) complex (PDB: 7JXU) structure was superposed with the MLKL(grey):RIPK3 (wheat) complex.

We thank the reviewers for their constructive comments and positivity towards our study. We have revised our manuscript extensively to accommodate the reviewers' suggestions (changes highlighted in main text with grey shading). Additionally, we have added Figure 5a plus additional and revised panels to the western blots in Figure 5, and additional data to Table 1 and Supplementary Table 1. We address each reviewer query (*blue italics*) with point-by-point responses (black, plain text) below.

REVIEWER COMMENTS

Reviewer #1 (Remarks to the Author):

Necroptosis, a regulated form of cell death, is executed by activating RIPK3 and MLKL through phosphorylation of RIPK3. However, the detailed molecular mechanisms of how RIPK3 activates MLKL are not fully understood yet. Moreover, There is an interspecies divergence of RIPK3 and MLKL; human RIPK3 activates human MLKL, whereas murine RIPK3 and murine MLKL. A previous study reported that crystal structures of murine RIPK3 and murine MLKL. In this manuscript, Meng et al. reports the crystal structure of the human RIPK3 kinase domain alone and in complex with the human MLKL pseudokinase domain. They find a distinct interface in the human RIPK3:MLKL complex, allowing the human RIP3/MLKL complex to remain inactive compared to murine one. Moreover, mutational analysis validated interface residues within the C-lobes of human RIPK3 and MLKL as crucial for necroptotic signaling in human cells.

Th present study elucidates the difference between human and murine RIPK3/MLKL complexes and provides a cue to investigate the mechanisms of how the inactive RIPK3/MLKL complex is activated. The reviewer requests several additional experiments and discussions to improve the manuscript. The followings are specific comments.

We thank Reviewer 1 for their critical evaluation of our study, and we absolutely agree with their interpretation of the significance of studying RIPK3:MLKL in the context of human orthologues.

Major points:

1. The authors propose a model that the RIPK3/MLKL-containing complex remains inactive in the cytosol before stimulation, and the complex becomes dissociated after stimulation. It is unclear whether free RIPK3 or free MLKL exists in the cells before or after stimulation. The authors need to show the stoichiometry of the RIPK3/MLKL complex, free RIPK3, and free MLKL in the cells. For example, Western blotting analysis using cell lysates before and after immuno-depletion of the RIPK3/MLKL complex with anti-RIPK3 or anti-MLKL antibodies may be feasible for evaluating the amounts of cellular pools of free RIPK3 or free MLKL.

We thank Reviewer 1 for raising an excellent point. We realise this is counter-dogma, which prompted us to examine the pre-association of RIPK3 and MLKL in human cells in some detail previously (reported recently in Garnish, Meng, Koide *et al. Nature Commun* 2021). The challenges with the experiment proposed by the reviewer are that suitable reagents for complete immunoprecipitation in cells are not available (as detailed in Samson *et al. CDD* 2021). One possibility is that the epitopes recognised by available monoclonal antibodies may well be buried by the interaction between RIPK3 and MLKL. Indeed, we used such a characteristic to our advantage in our recent report (Garnish *et al.* 2021; ref. 45 in this submission). By generating synthetic protein ligands termed Monobodies, which bind to the human MLKL

pseudokinase domain, we identified Monobody-27, which only recognised MLKL following its activation and disengagement from RIPK3, strongly arguing that MLKL is sequestered away in cells via RIPK3 complexation prior to receipt of a necroptotic cue. In contrast, Monobody-32, which binds on the opposite face of the MLKL pseudokinase domain, immunoprecipitated MLKL in complex with RIPK3 prior to exposure of cells to a necroptotic stimulus, validating the preassociation of these two proteins prior to necroptotic signaling. These findings were generated by studying the endogenous necroptosis machinery in HT29 cells, and are supported by the earlier work of Sun *et al.* (*Cell* 2012; ref. 4 in the manuscript). In the Sun *et al.* study, co-expressed exogenous human MLKL and human RIPK3 were reported to co-immunoprecipitate, consistent with the notion that these two proteins occur in a stable complex prior to induction of necroptosis.

Based on the reviewer's query, we have elaborated on the rationale for our claims in the revised Discussion (pages 13-14, lines 366-374), which build on our recent studies of the endogenous RIPK3:MLKL interaction in human cells in and are validated by our structural studies in the present manuscript.

2. Lines 104-111; The authors claim that human RIPK3 S227 phosphorylation is required for the interaction of MLKL. Assuming that S227 phosphorylation is mediated by RIPK3 itself, RIPK3 must be activated when generated in Sf9 cells. However, T357 and S358 of MLKL are not phosphorylated in the complex. The authors mention that "human MLKL T357 and S358, are distal from the catalytic HRD motif of RIPK3 (Supp. Fig. 3f), and a conformational change would be required for their phosphorylation to occur" (lines 259-261). In contrast, a previous study reported that both murine RIP3 and MLKL are phosphorylated in the complex (Cell Reports 5, 70-78, 2013). Considering the distance between the activation loop of murine MLKL and the catalytic HRD motif of murine RIPK3, the authors need to explain the difference in more detail.

We thank Reviewer 1 for raising another important point; we absolutely agree the role of phosphorylation in human RIPK3-MLKL complex assembly is of enormous interest. We unambiguously observed no electron density to support phosphorylation of human MLKL T357 or S358 in our complex structure reported in this study. While we apologize if the reviewer has more knowledge of the mouse complex than us, our reading of the Xie *et al.* 2013 paper mentioned by the reviewer is that there was no evidence for mouse MLKL phosphorylation within their crystal structure, where the RIPK3 substrate residue, mouse MLKL S345, was not modelled owing to a lack of electron density. Our interpretation of their paper is that mouse RIPK3 was similarly co-expressed in complex with mouse MLKL, and MLKL S345 could subsequently be phosphorylated in *in vitro* kinase reactions. If our interpretation is correct, this would parallel what we saw with the human RIPK3-MLKL complex, and reported in Garnish *et al.*, *Nature Commun* (Figure 1f therein) recently.

To decisively address the reviewer's query regarding the evidence for the importance of human RIPK3 S227 in MLKL complex formation, we have now co-expressed the wild-type human MLKL pseudokinase domain and human RIPK3 kinase domain harboring either a S227A or D142N mutation in *Sf21* insect cells. The mutation of RIPK3 S227, or prevention of its autophosphorylation by introducing the D142N kinase-dead mutation, would ablate S227 phosphorylation, which was predicted to be crucial for MLKL engagement. As we now show in **Figure 5a** of our revised manuscript, either of these RIPK3 mutations prevented complex formation with MLKL, which unequivocally supports the role of RIPK3 S227

autophosphorylation in MLKL complex formation. We thank the reviewer for prompting further investigation on this point; we now describe these findings on page 11, lines 283-294.

We also take the reviewer's point that our Discussion of the activation loop disposition in the mouse complex of Xie *et al.* could be more detailed and have now elaborated on page 14, lines 380-385. The key point we have sought to underline is that the RIPK3-bound form of human MLKL adopts an open conformation with a helical activation loop, while the mouse counterpart is largely unstructured. Thus to achieve the same flexibility and access to the RIPK3 active site, the human MLKL activation loop would be required to undergo a substantial conformation change. We also now note that in a cellular context, in the necrosome, MLKL would be part of a larger cluster with RIPK3, which would mean that RIPK3 could act *in trans* upon adjacent subcomplexes.

3. Lines 150-162; The authors mention that K230 in the b3-strand in the N-lobe is proximal to Q356 in the activation loop helix in the open conformer of human MLKL. The authors previously reported that a similar proximal interaction of K219 and Q343 is detected in murine MLKL, and mutations of these residues result in a constitutive active mutant of MLKL (Immunity 39, 443-453, 2013). Given that topology of the aC helix and the activation loop between murine and human MLKL is different, it would be interesting to test whether mutation of these amino acids (K230 and Q356) of human MLKL may affect MLKL' function.

This is an excellent point. Because we have observed this open conformer in a monobody-bound human MLKL crystal structure (PDB 7JXU), we previously tested these mutations in our earlier publication (Garnish *et al.* 2021; Supplementary Fig. 2j). Induction of K230M or Q356A did not trigger constitutive activation, consistent with differences between the activation mechanisms of human and mouse MLKL. We have ensured these findings are clear in our revised version, with the inclusion of additional text on page 7, lines 170-172.

3. In Figure 5c, d, e, the signals of pMLKL are too weak. The authors need to change them to more representative ones. Moreover, the authors should show pRIPK3 as well as total RIPK3.

Our experience is that phosphorylated MLKL is much more difficult to detect than total MLKL in immunoblots, which may arise from a combination of the sensitivity of available antibodies and the abundance of pMLKL in cells. However, in light of the reviewer's comments, we have increased the exposure in the pMLKL blots or replaced panels with others from different replicates to improve the signal intensity. Additionally, as requested by the reviewer, we have included pRIPK3 blots. These data are presented in revised Figure 5d-f.

4. In Figure 5e, it is unclear why RIPK3 S227A cells do not undergo necroptosis even though phosphorylation of MLKL is induced. The authors mention that "such a transient interaction and phosphorylation of MLKL are not sufficient to facilitate necroptotic cell death" (lines 415-419). If the authors claim "transient interaction and phosphorylation of MLKL", the authors should investigate the kinetics of TSI-induced phosphorylation of MLKL and quantify the signaling intensities of pMLKL in RIPK3 S227A cells compared to RIPK WT cells following TSI stimulation.

We agree with the reviewer that this was an unexpected finding. We take the reviewer's point that a comparison of the kinetics of this phosphorylation event may identify some differences in magnitude of MLKL phosphorylation arising from wild-type vs S227A RIPK3, but there are confounding factors that prevent a direct comparison. The main challenge is that S227A

RIPK3 expression is consistently lower than that of wild-type RIPK3 (Figure 5f), which would be expected to result in a lower abundance of pMLKL in S227A RIPK3-expressing cells. While the differences in their abundance limits our capacity to directly compare the kinetics by which they phosphorylate MLKL, we performed a time course as recommended by the reviewer (Response Figure 1). This time course shows that RIPK3 S227A phosphorylated MLKL at an attenuated level, and the onset of MLKL phosphorylation is at 4.5 hours post-TSI stimulation, consistent to the end point results presented in revised Figure 5f. In contrast, wild-type RIPK3 induced some MLKL phosphorylation in the unstimulated, basal state, which likely reflects the higher abundance of wild-type RIPK3 relative to that of the S227A mutant. We have included these data here for the reviewer's benefit and, upon publication, readers will be able to access these data in the peer review file, but because of the lower mutant RIPK3 expression level, we do not feel we can comment in a meaningful manner on the kinetics in the main text.

Response Figure 1

Regardless of the kinetics of MLKL phosphorylation by S227A RIPK3, we do not observe cell death in IncuCyte cell death assays, which are monitored over a 24-hour timecourse, which we feel logically point to deficiencies in MLKL recruitment to the necrosome as the basis for compromised cell death in S227A RIPK3-expressing cells. We have sought to clarify our reasoning on page 13, lines 342-345 to ensure the basis for our conclusions are clear. The experiments performed in light of the reviewer's comment 2 above, which queried the role of RIPK3 S227 in MLKL engagement, prove extremely useful in this regard. Because recombinant S227A RIPK3 did not form a stable complex with MLKL when co-expressed in Sf21 cells, we are confident there are deficits in MLKL recruitment, which we now discuss as important support for our model on page 14, lines 374-377.

5. In Figure 5e, the expression of RIPK3 I209R is very low following dox treatment. If the expression levels of RIPK3 I209R are very low, it may be unfair to discuss whether this RIPK3 mutant is functional in terms of necroptosis induction. The authors need to discuss this point, alternatively change the Figures to more representative ones.

We thank the reviewer for pointing this out; we agree that RIPK3 I209R expression is lower than the other RIPK3 mutants, and is consistently so in other repeats. Since I209 resides in a

hydrophobic pocket of RIPK3, introduction of a positive charge could have destabilized this variant. We have updated text on page 12, lines 319-321 and 335-338 to clarify this. Nonetheless, we are confident I209R RIPK3 is likely to be defective in necroptotic signaling, because all 5 hydrophobic mutants in the vicinity consistently abolished necroptosis, supporting the importance of this hydrophobic interface in RIPK3-mediated signaling.

Minor point:

1. Line 254, P36 should be F36.

Corrected with thanks.

2. Line 297, there is no data of F386A mutant in Figure 5. The authors need to include it.

We have previously reported that the F386A substitution in human MLKL completely abrogates signaling function (Ref. 11, Figure 3). Because the F386A MLKL behaved comparably to the T357E/S358E mutant, we chose to include only one such loss-of-function mutant in this study as a control. We have now included further description of our reasoning and refer more clearly to the previous study of F386A MLKL on page 12, lines 313-315.

3. In Supplementary Figure 3, Figure 3e should be labeled Figure 3f. Figure 3e is missing.

Thanks for pointing this out. We misnumbered the legend, and this has now been corrected with thanks.

4. In Supplementary Figure 4a, TSEE should be spelled as T357E/S358E.

5. In Supplementary Figure 4d, I109R should be changed to I209R.

We thank the reviewer again for their careful reading of the manuscript. We have now corrected these errors. We have also corrected the label on this figure to Supplementary Figure 5.

Reviewer #2 (Remarks to the Author):

Review

The manuscript presents the crystal structures of human RIPK3 kinase domain alone and in complex with the MLKL pseudokinase domain. These are critical protein players in necroptotic cell death. MLKL is the terminal effector in necroptosis signaling pathway, and is recruited to RIPK3 and subsequently phosphorylated by RIPK3. Importantly, there are sequence and functional variability in the RIPK3/MLKL pairs between species, which is why determination of the structures of the human proteins is an important step toward the mechanistic understanding how this pathway is regulated.

The study is presented well, and it is satisfying that the authors put the findings in context of previous work in the field so as to highlight its own importance. Without this, it may seem that the structural finding is a mere incremental improvement over the previously determined structure of the protein complex from mouse especially since the relative overall orientation of the two kinase domains is similar. However, determination of the human protein complex should provide a definitive molecular-level viewpoint for future mechanistic studies of this important pathway.

In general, the structural studies of the RIPK3:MLKL complex are satisfactory, with some specific questions raised below:

1. Was autobuild attempted to model this structure? At 2.2 Å resolution, it is possible that autobuild is achievable, which should remove model bias toward the original molecular replacement models.

2. Related, how have the authors ensured that model bias has not been introduced with respect to the relative kinase domain lobe orientations (active versus inactive). Was molecular replacement or rigid body refinement conducted with the individual lobes?

These are excellent suggestions to address model bias. We did not use an autobuild to build our models, but instead took a different approach to eliminate model bias. First, we tried several models of both open (human MLKL in PDB 7JXU, and the mouse MLKL model from PDB 4M69 that was truncated by CHAINSAW) and closed conformation (human MLKL in PDB 4MWI) as search models for molecular replacement, and proceeded with the solution with the best statistics and density match. For example, molecular replacement with the closed conformer of MLKL (PDB 4MWI) resulted in Top LLG = 822.292 and Top TFZ = 26.4; whereas when the open conformer of MLKL (PDB 7JXU) was used as search model, this resulted in Top LLG = 1451.730 and Top TFZ = 21.8. The closed form also resulted in substantial density mismatch in the model, especially near the α C helix and the activation loop helix of MLKL and, accordingly, the open conformer solution was selected as a starting model. During refinement, we also performed simulated annealing to avoid model bias. This has been clarified in the revised Methods on page 21, lines 570-571.

In addition, in light of Reviewer 2's query, we have now performed an autobuild, which we present here. The initial autobuild solution resulted in near identical conformation to 7MON (Response Figure 2; RMSD = 0.339 over 408 Ca atoms across both chains). As a result, we are confident that the model initially presented does not reflect model bias. We have updated the methods to ensure our initial approach is clear to the reader (on page 21, lines 565-568).

The structure of the open conformation of RIPK3 alone (in complex with an inhibitor) is also presented. I have some reservations of this structure, namely the accompanying statistics and data treatment, in its current form. However, I hope that the authors will be able to address

most of my questions directly in rebuttal and with a few additions/changes to the structure and/or manuscript text.

The authors state that the diffraction data for the RIPK3/GSK483 structure suffered from anisotropy, and so ellipsoidal truncation was performed automatically in autoPROC with STARANISO. This is an acceptable treatment of the data; however, a few details could be provided which would further help explain this anisotropy effect and ensure that the data processing is appropriate. I would look forward to re-examining this structure if the following points could be addressed.

We would like to thank reviewer 2 for their insightful questions, which have allowed us to further explain the elliptical truncation methodology and how it compares with conventional approaches. As the reviewer will have noted, obtaining suitably diffracting crystals to solve the human RIPK3:GSK'843 structure was enormously challenging. This is reflected in ours being the first human RIPK3 kinase domain structure since it was first implicated in the necroptosis pathway 12 years ago. We are fortunate that elliptical truncation allowed us to get the most out of the dataset, despite the modest resolution of 3.2 Å.

Major questions:

1. How was the resolution limit reached? Did the authors carefully examine a F/σ vs resolution plot for each of the three principal axes to reach the resolution limit?

The resolution limit for elliptical truncation is determined automatically by the STARANISO module in autoPROC. It uses a threshold $I/\sigma I$ value greater than 1.2 to determine the cutoff in each axis. The $I/\sigma I$ value for our highest resolution shell in the elliptical data is 1.5. The CC1/2 value is 0.451.

Related, what was the F/σ of the discarded reflections?

To our knowledge, the autoPROC log files do not calculate the $I/\sigma I$ for the discarded reflections. However, the dataset processed spherically to a 3.23 Å cutoff has very low $I/\sigma I$ values below ~3.9 Å, which leads us to believe the $I/\sigma I$ value of the discarded reflections must be very low. The full statistics and table 1 (produced by STARANISO) for the spherically processed data are included below.

Resolution	#uniq	#Rfac	Rmerge	Rmeas	Rpim	#lsig	I/sigI	Compl. spherical		Multiplicity		CC(1/2)	#CCano	CC(ano)	SigAno	Compl. ellipsoidal	
								all	ano	all	ano					all	ano
80.759 - 8.762	1256	7175	0.082	0.091	0.038	1256	6.362	0.9976	1.0000	5.72	3.33	0.980	868	-0.1961	0.398	0.9976	1.0000
8.762 - 6.956	1197	7613	0.168	0.184	0.073	1197	4.990	1.0000	1.0000	6.36	3.52	0.9881	952	-0.1316	0.631	1.0000	1.0000
6.956 - 6.077	1171	7863	0.349	0.379	0.146	1171	3.447	1.0000	1.0000	6.71	3.64	0.9649	974	-0.0381	0.755	1.0000	1.0000
6.077 - 5.521	1172	8014	0.484	0.525	0.200	1172	2.744	1.0000	0.9990	6.84	3.68	0.9452	1001	-0.0655	0.771	1.0000	0.9990
5.521 - 5.126	1151	6901	0.477	0.523	0.212	1151	2.388	1.0000	0.9980	6.00	3.23	0.9611	964	-0.0834	0.738	1.0000	0.9989
5.126 - 4.823	1174	6867	0.508	0.558	0.229	1174	2.368	1.0000	0.9981	5.85	3.12	0.9502	974	-0.0374	0.775	1.0000	0.9988
4.823 - 4.582	1123	7140	0.597	0.651	0.257	1123	2.224	1.0000	0.9980	6.36	3.39	0.9358	956	-0.0695	0.786	1.0000	0.9971
4.582 - 4.382	1144	7413	0.719	0.782	0.305	1144	1.924	1.0000	0.9971	6.48	3.44	0.9217	986	0.0436	0.762	1.0000	0.9953
4.382 - 4.214	1140	7521	0.854	0.930	0.363	1140	1.626	1.0000	0.9990	6.60	3.49	0.9037	983	-0.0260	0.779	1.0000	0.9982
4.214 - 4.068	1170	7826	1.057	1.148	0.444	1170	1.446	1.0000	1.0000	6.69	3.54	0.8096	1015	0.0336	0.760	1.0000	1.0000
4.068 - 3.941	1138	7690	1.451	1.573	0.603	1138	1.153	1.0000	0.9990	6.76	3.56	0.6645	1001	0.0606	0.793	1.0000	0.9977
3.941 - 3.829	1121	7677	1.712	1.855	0.708	1121	0.981	1.0000	0.9970	6.85	3.59	0.6911	996	-0.1322	0.720	1.0000	0.9921
3.829 - 3.728	1142	7846	2.200	2.383	0.906	1142	0.789	0.9991	0.9981	6.87	3.61	0.5809	1009	-0.0408	0.750	0.9974	0.9938
3.728 - 3.637	1132	7400	2.610	2.838	1.104	1132	0.658	1.0000	0.9980	6.54	3.44	0.5647	999	0.0346	0.784	1.0000	0.9918
3.637 - 3.554	1134	7014	3.244	3.547	1.421	1134	0.500	0.9991	0.9981	6.19	3.24	0.4016	998	0.0424	0.777	0.9954	0.9892
3.554 - 3.478	1150	7060	5.209	5.701	2.292	1150	0.332	1.0000	0.9990	6.14	3.22	0.1621	992	0.0475	0.750	1.0000	0.9934
3.478 - 3.409	1120	6631	6.007	6.595	2.697	1120	0.271	1.0000	0.9932	5.92	3.10	0.2411	968	0.0041	0.742	1.0000	0.9911
3.409 - 3.345	1127	7087	7.445	8.125	3.224	1127	0.229	1.0000	0.9990	6.29	3.29	0.1493	989	-0.0221	0.721	1.0000	1.0000
3.345 - 3.285	1116	7144	7.368	8.031	3.166	1116	0.227	1.0000	0.9971	6.40	3.34	0.1206	975	-0.0075	0.756	1.0000	1.0000
3.285 - 3.229	1143	7408	12.309	13.388	5.218	1143	0.154	1.0000	1.0000	6.48	3.39	0.0756	1011	-0.0633	0.725	1.0000	1.0000
Total: 80.759 - 3.229	23021	147290	0.581	0.634	0.250	23021	1.780	1.0000	0.9985	6.40	3.41	0.9701	19611	-0.0322	0.736	0.9997	0.9981

	Overall	InnerShell	OuterShell
Low resolution limit	80.759	80.759	3.285
High resolution limit	3.229	8.762	3.229
Rmerge (all I+ & I-)	0.581	0.082	12.309
Rmerge (within I+/I-)	0.572	0.077	12.627
Rmeas (all I+ & I-)	0.634	0.091	13.388
Rmeas (within I+/I-)	0.681	0.092	15.035
Rpim (all I+ & I-)	0.250	0.038	5.218
Rpim (within I+/I-)	0.365	0.049	8.078
Total number of observations	147324	7182	7411
Total number unique	23021	1256	1143
Mean(I)/ sd (I)	1.8	6.4	0.2
Completeness (spherical)	100.0	99.8	100.0
Completeness (ellipsoidal)	100.0	99.8	100.0
Multiplicity	6.4	5.7	6.5
CC(1/2)	0.970	0.980	0.076
Anomalous completeness (spherical)	99.9	100.0	100.0
Anomalous completeness (ellipsoidal)	99.8	100.0	100.0
Anomalous multiplicity	3.4	3.3	3.4
CC(ano)	-0.032	-0.196	-0.063
DANO / sd (DANO)	0.736	0.398	0.725

We used the Diffraction Anisotropy Server (as suggested in query 6) to obtain an F/sig value of 2.26 for discarded reflections on a similarly elliptically truncated dataset. We inputted the scaled and merged, but untruncated, data from autoPROC, and manually set the resolution cutoff for each axis to be the same as the STARANISO data (5.7, 3.7 and 3.1 Å in a* b* and c* respectively). It is worth noting that the server did not truncate the data in exactly the same fashion as STARANISO, but the two were comparable: 11832 reflections remained in the Diffraction Anisotropy Server data and 10790 were included in the STARANISO processed data.

2. I have concerns with the low completeness of even the ellipsoidal truncated data (90.7%, 56.9%). Does the completeness improve if a slightly lower (poorer) resolution cutoff is chosen (i.e. 3.3 Å vs 3.2 Å)?

This is a very important point and we are grateful to the reviewer for prompting further justification of the ellipsoidal truncation approach. We note that the completeness is high in the lowest (poor) resolution shells and is only low in the highest resolution shells, as is typical of this method (as described further below). The completeness over each resolution shell is shown below (from STARANISO).

To test whether completeness improved with a lower resolution cutoff, we used the STARANISO online server (<https://staraniso.globalphasing.org>; rather than autoPROC) to reprocess the data using an I/sigI threshold of 1.8. This resulted in a dataset cut to 3.4 Å in the highest resolution axis. Interestingly, most of the statistics, including the completeness were not improved at this lower resolution cut off, although the I/sigI improved considerably as would be expected. The STARANISO outputs from these two analyses are presented below for the reviewer's benefit.

Resolution	#uniq	#Rfac	Rmerge	Rmeas	Rpim	#lsig	I/sigI	Compl. spherical		Multiplicity		CC(1/2)	#CCAno	CC(ano)	SigAno	Compl. ellipsoidal	
								all	ano	all	ano					all	ano
80.759 - 11.781	540	3205	0.078	0.086	0.036	540	6.711	0.9945	1.0000	5.94	3.66	0.9639	344	-0.1164	0.334	0.9945	1.0000
11.781 - 9.259	539	2938	0.081	0.089	0.037	539	6.217	1.0000	1.0000	5.45	3.10	0.9951	382	-0.1814	0.408	1.0000	1.0000
9.259 - 8.037	540	3250	0.123	0.136	0.055	540	5.642	1.0000	1.0000	6.02	3.36	0.9923	420	-0.243	0.561	1.0000	1.0000
8.037 - 7.285	539	3464	0.177	0.193	0.076	539	5.062	1.0000	1.0000	6.43	3.54	0.985	433	-0.065	0.642	1.0000	1.0000
7.285 - 6.730	538	3538	0.276	0.3	0.117	538	3.974	1.0000	1.0000	6.58	3.58	0.9773	442	-0.2075	0.713	1.0000	1.0000
6.730 - 6.319	542	3623	0.336	0.365	0.14	542	3.491	1.0000	1.0000	6.68	3.64	0.9675	450	-0.0068	0.758	1.0000	1.0000
6.319 - 5.983	540	3708	0.375	0.406	0.155	540	3.248	0.9507	0.9669	6.87	3.69	0.9625	459	-0.0927	0.726	0.9507	0.9669
5.983 - 5.679	538	3672	0.434	0.47	0.179	538	3.142	0.8636	0.8738	6.83	3.67	0.9366	461	-0.0465	0.748	0.8636	0.8738
5.679 - 5.408	538	3548	0.436	0.474	0.184	538	2.835	0.8018	0.8107	6.59	3.52	0.9653	464	-0.1016	0.759	0.8499	0.8517
5.408 - 5.169	541	3286	0.404	0.443	0.179	541	2.967	0.7462	0.756	6.08	3.24	0.9557	458	-0.1038	0.708	0.8601	0.8571
5.169 - 4.957	540	3167	0.369	0.405	0.165	540	3.084	0.7317	0.7376	5.87	3.12	0.9557	455	-0.0583	0.756	0.9045	0.903
4.957 - 4.774	539	3311	0.443	0.486	0.196	539	2.965	0.7046	0.7078	6.15	3.26	0.9586	461	-0.0836	0.756	0.9374	0.9359
4.774 - 4.603	540	3389	0.446	0.487	0.194	540	2.944	0.6601	0.663	6.28	3.34	0.9492	455	-0.0836	0.815	0.9571	0.9556
4.603 - 4.449	540	3385	0.495	0.54	0.214	540	2.718	0.6265	0.6265	6.27	3.34	0.9364	459	0.0319	0.782	0.9654	0.963
4.449 - 4.304	539	3413	0.507	0.555	0.221	539	2.623	0.5859	0.583	6.33	3.37	0.9163	453	-0.0292	0.74	0.9779	0.9769
4.304 - 4.160	538	3421	0.552	0.604	0.242	538	2.558	0.5275	0.5225	6.36	3.36	0.8854	453	-0.0037	0.799	0.9689	0.9689
4.160 - 4.017	542	3446	0.682	0.744	0.294	542	2.263	0.4597	0.4557	6.36	3.37	0.7635	462	0.0135	0.789	0.9517	0.9594
4.017 - 3.866	537	3380	0.748	0.818	0.326	537	2.099	0.3698	0.3668	6.29	3.32	0.7472	457	-0.0826	0.814	0.8612	0.8736
3.866 - 3.697	540	3326	0.79	0.865	0.348	540	1.973	0.2839	0.2731	6.16	3.30	0.7172	438	-0.0956	0.785	0.8293	0.8371
3.697 - 3.229	540	3019	1.031	1.142	0.484	540	1.506	0.0717	0.0636	5.59	3.07	0.451	407	0.0423	0.891	0.5689	0.557
Total: 80.759 - 3.229	10790	67489	0.259	0.283	0.113	10790	3.401	0.4687	0.4511	6.26	3.39	0.9788	8813	-0.0767	0.722	0.9075	0.908

STARANISO TABLE 1 – original processing

Number of active ice-rings within this resolution range = 0

Criteria used in determination of diffraction limits:

local(I/sigI) >= 1.20

	Overall	InnerShell	OuterShell
Low resolution limit	80.759	80.759	3.697
High resolution limit	3.229	11.781	3.229
Rmerge (all I+ & I-)	0.259	0.078	1.031
Rmerge (within I+/I-)	0.250	0.073	0.931
Rmeas (all I+ & I-)	0.283	0.086	1.142
Rmeas (within I+/I-)	0.297	0.085	1.131
Rpim (all I+ & I-)	0.113	0.036	0.484
Rpim (within I+/I-)	0.160	0.044	0.634
Total number of observations	67513	3210	3021
Total number unique	10790	540	540
Mean(I)/sd(I)	3.4	6.7	1.5
Completeness (spherical)	46.9	99.4	7.2
Completeness (ellipsoidal)	90.7	99.4	56.9
Multiplicity	6.3	5.9	5.6
CC(1/2)	0.979	0.964	0.451
Anomalous completeness (spherical)	45.1	100.0	6.4
Anomalous completeness (ellipsoidal)	90.8	100.0	55.7
Anomalous multiplicity	3.4	3.7	3.1
CC(ano)	-0.077	-0.116	0.042
DANO /sd(DANO)	0.722	0.334	0.891

STARANISO TABLE 1 – higher threshold processing

Number of active ice-rings within this resolution range = 0

Criteria used in determination of resolution cut:

Rpim <= 0.6000
I/sigI >= 2.00
CC(1/2) >= 0.3000

	Overall	InnerShell	OuterShell
Low resolution limit	80.759	80.759	4.081
High resolution limit	3.420	10.444	3.420
Rmerge (all I+ & I-)	0.292	0.141	2.237
Rmerge (within I+/I-)	0.286	0.112	2.413
Rmeas (all I+ & I-)	0.319	0.156	2.408
Rmeas (within I+/I-)	0.337	0.133	2.806
Rpim (all I+ & I-)	0.126	0.066	0.885
Rpim (within I+/I-)	0.178	0.071	1.428
Total number of observations	52222	4562	4537
Total number unique	8366	764	752
Mean(I)/sd(I)	5.8	15.3	3.0
Completeness (spherical)	43.0	99.9	9.6
Completeness (ellipsoidal)	87.3	99.9	57.1
Multiplicity	6.2	6.0	6.0
CC(1/2)	0.892	0.764	0.672
Anomalous completeness (spherical)	41.1	99.8	8.6
Anomalous completeness (ellipsoidal)	87.2	99.8	56.0
Anomalous multiplicity	3.4	3.6	3.3
CC(ano)	-0.003	0.058	-0.066
DANO /sd(DANO)	0.884	1.193	1.005

Or can the authors provide rationale (with references, if applicable) that it is acceptable for anisotropic and truncated data to have such low completeness? In general, the quality of an electron density map can suffer greatly by low completeness. Is the same true for anisotropic data?

We agree with the reviewer that this is not a common treatment of data, however it is clear from the literature that this a well-established method for cases where data are severely anisotropic. In these cases, elliptical corrections led to completeness comparable to what we observed. There are many published examples, some of which are listed here:

Youn *et al.*, *Sci Rep* (2017) 7:2595 “Construction of novel repeat proteins with rigid and predictable structures using a shared helix method”

Kumar *et al.*, *Sci Rep* (2017) 7:14288 “Combined x-ray crystallography and computational modelling approach to investigate the Hsp90 C-terminal peptide binding to FKBP51”

Duda *et al.*, *Structure* (2013) 21:1030-1041 “Structure of HHARI, a RING-IBR-RING Ubiquitin Ligase: Autoinhibition of an Ariadne-Family E3 and Insights into Ligation Mechanism”

Branscum *et al.*, *Protein Science* (2019) 28:2099-2111 “Insights revealed by the co-crystal structure of the *Saccharomyces cerevisiae* histidine phosphotransfer protein Ypd1 and the receiver domain of its downstream response regulator Ssk1”

Additionally, there is an excellent explanation of the approach, its rationale and illustration of its utility in Robert *et al.*, *Sci Rep* (2017) 7:17013 “X-ray diffraction reveals the intrinsic difference in the physical properties of membrane and soluble proteins”.

We have ensured that the rationale for this data treatment is now presented in the Methods on page 20, lines 574-577, along with further details that others typically present when using this approach here and in the revised Table 1.

3. Related to resolution limit and completeness, have the authors examined electron density maps calculated at lower resolution limits? Can better maps be obtained by different treatment of the data? The maps as presented in Fig S4 are quite poor even for 3.2 Å resolution. The resolution limit should not be pushed unnecessarily especially if the map quality is better achieved at lower resolution cutoff.

This is a good question. We initially solved this dataset by processing it in DIALS to 3.55 Å, with a spherical cut off. While the CC1/2 was still significant at this cutoff, and data were 100% complete in all shells, the I/sigI was very poor (0.4 in highest resolution shell). The maps were very poor and it was difficult to determine sidechain positions. When reprocessed with autoPROC and solved with the elliptically truncated data, the maps were immediately better, although not fantastic, but we could place sidechains with more confidence. For that reason, we proceeded with the elliptically truncated data for model building and refinement.

Included below are some sample images from the $2Fo-Fc$ map post first refinement of the 3.55 Å DIALS processed dataset (purple map, green model) vs the $2Fo-Fc$ map post first refinement with the STARANISO processed data (blue map, yellow model). The data were solved with the same molecular replacement model and refined with the same conditions in Phenix.refine. Helices and sidechains are better resolved in the elliptically truncated data, despite its lower completeness.

Maps are both contoured to 1σ . Images taken from Coot.

To further investigate the reviewer's question of whether reducing resolution might lead to improved maps, we have reprocessed the data using the STARANISO server (<https://staraniso.globalphasing.org/>), which provides a greater capacity to adjust settings. We selected a higher $I/\sigma I$ cutoff (1.8) to impose a more conservative elliptical truncation, which resulted in a 3.42 Å highest resolution dataset, with an $I/\sigma I$ of 3.0 in the highest resolution shell. Importantly, this did not yield improved merging statistics compared to the original data and the arising maps were worse than with the stricter truncation. We transferred the R_{free} flags from our original 3.22 Å elliptically-truncated dataset and used the same initial model as we started with for the original 3.22 Å dataset to phase the data in PHASER, followed by iterative model building and refinement in Coot and Phenix, analogous to our approach using the 3.22 Å dataset. We examined the quality of the maps throughout refinement and once R_{free} had reached a similar value to the structure presented in the paper, compared the maps from each structure. While some regions of the map are comparable between the 3.42 and 3.22 Å processed data, other regions have very poor density in the 3.42 Å maps, particularly the N-lobes of chains C and D, that would preclude confident fitting of sidechains. As a result, we concluded that the 3.22 Å elliptically-truncated dataset was superior to that with the 3.42 Å cutoff in terms of electron density map quality and their amenability to model building.

We have included below some sample images of the maps from these two structures. The R_{work} and R_{free} of the 3.22 Å final structure are 0.2394/0.2855. The R_{work} and R_{free} for the 3.42 Å structure are 0.2370/0.2983. This was the best R_{free} achieved for the 3.42 Å structure, which we think further validates that the 3.22 Å cutoff gives the best maps, because a better agreement between the data and the model (lower R_{free}) could be achieved with the 3.22 Å processing.

R-work: 0.2394, R-free: 0.2855

R-work: 0.2370 R-free: 0.2983

Comparing 3.22 Å and 3.42 Å angstrom maps

3.22 Å maps are in blue ($2Fo-Fc$), the corresponding model is in green (carbons). 3.42 Å maps are in purple ($2Fo-Fc$), the corresponding model is in blue (carbons). The white arrows indicate regions where we think the 3.22 Å maps are of better quality and allow side chains to be better modelled. Maps are both contoured to 1σ . Images taken from Coot.

4. Similar to questions about the RIPK3:MLKL complex structure, was molecular replacement and/or rigid body refinement conducted with the individual kinase domain lobes to ensure that their relative orientation (active vs inactive) has not suffered from model bias, especially here at lower resolution and with anisotropic data?

We thank the reviewer for raising another important point. We usually use simulated annealing to address model bias, but when we attempted it with the human RIPK3:GSK'843 structure, the density was too poor, causing the Clashscore and Ramachandran outliers to increase markedly after refinements including simulate anneals (both Cartesian and torsion angle).

We thank the reviewer for the suggestion of solving the structure using the individual lobes as molecular replacement models. We separated the N- and C-lobes of the kinase model used as the initial molecular replacement model at the hinge region, and inputted them into PHASER as separate components. Phaser found one solution with LLG of 1732.53 and TFZ of 17.1. The resulting kinase domains in the molecular replacement solution align closely with the N- and C-lobe orientations within our final structure depicted in the two examples below, corresponding to two (of four) different chains in the human RIPK3:GSK'843 structure asymmetric unit. In the final structure all four chains have the same N-lobe:C-lobe orientation.

Our final structure is shown in pink for two different chains in the asymmetric unit, with the new molecular replacement solution using the split N- and C-lobes in various colours. Based on these data, we are convinced that the relative orientation of these domains is not influenced by model bias.

5. Are the Rmerge and I/sigma values in Table 1 for 7MX3 from after truncation? Related, can the authors provide Rmerge, completeness, and I/sigma values for both before and after anisotropic truncation?

The original statistics were for the truncated data, but we have now updated Table 1 with the statistics for the data processed spherically to 3.23 Å. These data are also included for the reviewer in the response to their first query. We agree that presenting these data in Table 1 is a good idea, and important for the reader's information.

6. The authors could run an orthogonal analysis of their original scaled reflection file to validate the choice of resolution cutoff. Namely, the UCLA-DOE LAB — Diffraction Anisotropy Server. This would help cross-validate the cutoff determined by STARANISO.

We thank the reviewer for this suggestion. Reassuringly, the Diffraction Anisotropy Server also identified the strong anisotropy that was reported by STARANISO and imposed a similar elliptical truncation to that of STARANISO. The Diffraction Anisotropy Server truncated the data to 6.0, 4.0 and 3.6 Å in the a*, b* and c* axes respectively, based on an F/sig cutoff of 3. The F/sig value of the discarded reflections in this instance was 2.37. The autoPROC/STARANISO cutoffs were 5.73, 3.72 and 3.14 in the a*, b* and c* axes, respectively. Inputting the STARANISO cutoff values into the Diffraction Anisotropy Server

resulted in an F/σ value of discarded reflections as 2.26, and this cutoff resulted in an extra 2693 reflections being included in the data. We have included the output from the server below for the reviewer's benefit, noting the concordant findings from both software.

Diffraction anisotropy server auto cut off

The recommended resolution limits along/near to a^* , b^* , c^* are
 6.0 Ang 4.0 Ang 3.6 Ang
 These are the resolutions at which F/σ drops below an arbitrary cutoff of 3.0

28566 reflections were in the initial data set. 19427 were discarded because they fell outside the specified ellipsoid with dimensions 1/ 6.0, 1/ 4.0, 1/ 3.6 Å along a^* , b^* , c^* , respectively. These discarded reflections had an average F/σ of 2.37. 9139 reflections remain after ellipsoidal truncation. Anisotropic scale factors were then applied to remove anisotropy from the data set. Lastly, an isotropic B of -85.90 Å^2 was applied to restore the magnitude of the high resolution reflections diminished by anisotropic scaling. The following pseudo precession images illustrate the individual steps.

Diffraction anisotropy server staraniso cut off

User input resolution limits along/near to a^* , b^* , c^* are
 5.7 Ang 3.7 Ang 3.1 Ang
 These are the resolutions at which F/σ drops below an arbitrary cutoff of 3.0

28566 reflections were in the initial data set. 16734 were discarded because they fell outside the specified ellipsoid with dimensions 1/ 5.7, 1/ 3.7, 1/ 3.1 Å along a^* , b^* , c^* , respectively. These discarded reflections had an average F/σ of 2.26. 11832 reflections remain after ellipsoidal truncation. Anisotropic scale factors were then applied to remove anisotropy from the data set. Lastly, an isotropic B of -77.78 Å^2 was applied to restore the magnitude of the high resolution reflections diminished by anisotropic scaling. The following pseudo precession images illustrate the individual steps.

7. Can the authors provide details as to how/when during processing the R_{free} reflections were chosen?

For RIPK3:MLKL structure (PDB: 7MON), R_{free} reflections were added using phenix.reflection_file_editor before molecular replacement on indexed and merged reflection data. For the RIPK3 GSK'843 structure (PDB: 7MX3), R_{free} flags were added using phenix.refine as the first step in the first refinement of the STARANISO processed data, post molecular replacement. They were added to 10% of the reflections, and the flag set is elliptical corresponding to the reflection data. We have now added further description of what we did to the Methods on page 20, lines 564 and 575-577.

8. The Clashscore and Ramachandran distribution could be improved.

We thank the reviewer for prompting us to revisit these important aspects of structure validation. Within the RIPK3:MLKL complex structure, we are comfortable with the Ramachandran distribution; there are no Ramachandran outliers in this model. Similarly, we are comfortable with the Clashscore for the RIPK3:MLKL complex structure, which is in line

with other 2.3 Å structures, as noted in the PDB deposition report. While we take the reviewer's point that the Clashscore for our RIPK3:GSK'843 structure is at face value high, it is attributable to the low resolution of this structure and, in our PDB deposition report, can be seen to be better than most structures of comparable resolution. Due to the poor maps in the RIPK3:GSK'843 structure it was very challenging to improve the Ramachandran distribution to its current state. We paid particular attention to the Ramachandran distribution while model building. After much work, we believe that the main chain is modelled correctly, with carbonyls and amides flipped correctly, consistent with the secondary structure elements of the structure. A slightly better Ramachandran distribution can be achieved while model building in Coot. However, because of the poor density, during refinement the mainchain drifts slightly from the ideal ϕ and ψ angles, resulting in the few Ramachandran unfavoured residues.

9. Does the disulfide bond between the alpha-G helices of apposing copies induce a conformational change? This is important, since the manuscript states that "the most substantial structural changes upon MLKL complexation occur on and around the alphaG helix", and because the disulfide forms between Cys residues on the alphaG helix.

This is an excellent question. We agree with the reviewer that this disulfide bond could contribute to the orientation we observed for the human RIPK3 α G helix, and we have now added further text on this possibility on page 9, lines 226-228. As a side note, we have attempted to crystallize the human RIPK3 kinase domain after mutating this Cys, but we were unable to generate diffracting crystals, so it seems this disulfide bond serendipitously contributes to its crystallizability.

Several additions to the SAXS data section should also be considered:

1. The authors should include one or more molecular weight determinations from the SAXS data, namely the Porod Volume, volume of correlation, etc. Is the molecular weight consistent with what is expected for the complex? This can be added to Table S1. These types of calculation are readily available in many SAXS software packages (e.g. RAW) and online servers (SAXSMoW).

We agree these are important analyses, and apologize for their omission in our first submission. We have now included a molecular weight estimate derived from Bayesian inference (<https://www.nature.com/articles/s41598-018-25355-2.pdf>) within the ATSAS package in Table S1; these estimates are consistent with the calculated complex molecular weight. The Bayesian inference output uses molecular weight estimates from four methods, including MoW, to derive a consensus estimate. Within this analysis, the MoW values for the apo complex is 69514, and for the Compound-10 bound complex is 76891 Da, which are slightly larger than the Bayesian consensus values, but are similarly consistent with the complexes containing one copy of each of RIPK3 and MLKL.

2. Based on the "good fit" of the theoretical curve to the crystal structure, it seems that calculation of a bead model and/or electron density (DENS) would make for a descriptive figure when superposed with the crystal structure. The authors might consider including such a model to Fig S1, though this is not absolutely required.

We realise these are commonly used to illustrate the molecular envelopes calculated from scattering data, although we typically avoid their inclusion whenever we present CRY SOL analyses. Because bead models and CRY SOL analyses are reinterpretations of the same data,

we feel this can often be misleading to readers unfamiliar with these types of analyses. Often, inclusion of both representations can convey the impression of multiple lines of support for a model, despite the origins in the same dataset.

Additional general questions/comments on manuscript:

1. Intro paragraph 1, lines 33-37, can this sentence be better clarified?

We agree this was a cumbersome tract of text. We have edited accordingly to break down the elements into a more palatable form.

2. Can the phosphorylation of RIPK3, and lack of phosphorylation of MLKL, verified in vitro?

This is an excellent point and was also of great interest to us considering the unambiguous lack of MLKL phosphorylation within the electron density for our complex crystal structure. Our earlier data indicated the role of RIPK3-mediated phosphorylation of MLKL in dissociating the RIPK3:MLKL complex (Figure 1f in Garnish *et al.* 2021). In these experiments, using an *in vitro* kinase assay, we found that recombinant His₆-tagged human MLKL:RIPK3 complex is dissolved when incubated with ATP and Mg²⁺ *in vitro*, which we attributed to RIPK3-mediated phosphorylation of MLKL causing complex dissociation. In addition, we now show in **Figure 5a** with accompany text (page 12, lines 283-294) that RIPK3 phosphorylation is a crucial determinant of the capacity for co-expressed RIPK3 and MLKL to assemble into a complex. By mutating the key phosphosite in RIPK3 (S227) or by introducing a kinase-dead mutation (D142N), we blocked formation of complexes with MLKL. While these data clearly illustrate that RIPK3 needs to be phosphorylated to engage MLKL, they do not address whether MLKL can be phosphorylated within the complex.

To more directly address this prospect, we performed PhosTag SDS-PAGE and immunoblot analysis on recombinant RIPK3:MLKL complex to examine the different phosphorylated species (Response Figure 3). While the distribution of RIPK3 phospho-species did not change markedly following exposure to conditions that enable kinase activity – incubation with ATP/Mg²⁺ – we observed two MLKL bands within the purified RIPK3:MLKL complex that coalesced into a single, more slowly migrating MLKL form following ATP/Mg²⁺ incubation. This suggests that there is likely a mixture of dephospho- and phospho-MLKL present in the recombinant complex. However, it remains a matter of ongoing interest to determine the nature of these phospho-forms and whether, for example, double phosphorylation of MLKL at T357 and S358 are the key cues for dissociation of MLKL from RIPK3. Because only dephosphorylated MLKL was observed in the RIPK3:MLKL crystal structure, we feel this provides additional support for the idea that MLKL phosphorylation drives disassembly of the complex, resulting in only dephospho-MLKL and RIPK3 co-crystallizing.

Response Figure 3

3. Supp Fig 1c difference density map, what is the contour level shown?

We apologize missing this detail and thank the reviewer for picking up this omission. The *2Fo-Fc* density was contoured at 1.0σ , as we have now mentioned in the legend.

4. Thermal shift assay: can another kinase as negative control be tested?

This is an excellent suggestion. We have performed thermal stability shift assays with each of mouse MLKL pseudokinase domain and an unrelated kinase, PK1 (Oliver *et al.* 2021, *Nat Comm* 12:1002), both of which we suspected might be suitable as negative controls (Response Figure 4). Compound-10 had no detectable effect on the thermal stability of either protein, which fits with our observation that only part of the compound is bound in the ATP-binding cleft of human MLKL. Importantly, these data indicate that while Compound-10 is promiscuous, there is still some selectivity in its targets.

Response Figure 4

5. Was AMPPNP tested instead of inhibitor for complex formation?

We have not tried AMPPNP as an agent to stabilize the complex for crystallization. We focused our efforts on using various ATP-competitive inhibitors because of their higher affinity for RIPK3, although in future it would be worth considering AMPPNP as a point of comparison.

6. Can the change between mouse and human (visualized in Fig 2c) be explained by the binding of compound 10?

This is an interesting idea, although our conclusion is that differences between mouse and human MLKL conformations within their respective RIPK3 complexes most likely arise from intrinsic structural differences rather than being driven by Compound-10 binding. We have previously reported another human MLKL structure in an open conformation captured by a Monobody (PDB: 7JXU) in the absence of Compound 10. The two structures exhibit near identical conformations (Response Figure 5, left) (RMSD = 0.655 over 261 C α atoms between 7MON chain A and 7JXU chain A). In contrast, this human MLKL open conformer also exhibits very different conformation from the mouse MLKL conformer in complex with RIPK3 (PDB: 4M69) (Response figure 5, right) (RMSD = 0.792 over 214 C α atoms), indicating that human MLKL exhibits intrinsic structural differences from mouse MLKL, which do not rely on Compound-10 binding. We now emphasize on page 7, lines 163-167, in light of the reviewer's query.

Response figure 5

Does compound 10 bind mouse MLKL?

We thank the reviewer for prompting further investigation on this. To test whether Compound 10 binds mouse MLKL, we performed a thermal stability shift assay on the mouse MLKL pseudokinase domain, as described in response to this reviewer's query #4 (Response Figure 4). Unsurprisingly, considering the unusual binding mode to human MLKL in the crystal structure and a binding event that likely arises from very high Compound-10 concentrations in our crystallization drops, we did not detect Compound-10 binding to mouse MLKL. These data

serve as a useful control and reinforce the idea that, while Compound-10 is promiscuous, it does exhibit some target selectivity.

Overall, the manuscript is well written and should be understandable to a broad audience. The questions raised in this review will need to be addressed before the entirety of the structural data and interpretation are acceptable for publication. However, based on what is presented in its current form, I hope and expect that the authors will be able to address the issues and that the biological implications of the work, and therefore the message of the manuscript as a whole, will largely be upheld.

We are grateful to Reviewer 2 for their generous appraisal of our study. We thank them for their positivity and helpful suggestions for improvement of the manuscript.

Reviewer #3 (Remarks to the Author):

In current work, the authors have determined the structure of human RIPK3 kinase domain alone and in complex with MLKL pseudokinase. Current complex structure was compared with mouse counterpart, the structure of mouse RIPK3-MLKL complex. The authors identified how human RIPK3 maintains MLKL in an inactive conformation prior to induction of necroptosis and residues involved in necroptotic signaling were identified and characterized using human cells.

Overall work has good quality. However, major revisions are needed before current manuscript be suitable for publication in current journal. I have following concerns in current manuscript

We thank the reviewer for their positive appraisal of the work, and appreciate their constructive suggestions for improving the manuscript.

-English grammar should be checked throughout the manuscript. Many paragraphs are not written clearly and do not explain the meaning of sentences. In single sentence, many results and interpretations are written and merged together with no clarity and grammar.

We appreciate the reviewer's careful reading of the manuscript and have ensured we have carefully read the revised version with these comments in mind.

-Human RIPK3 (residue 2-316; C3S, C110A)+GSK'843 complex structure was determined at low resolution (~3.23 Å) and compared with high resolution (~2.23 Å) structure of Human RIPK3 (residues 1-316; C3S, C110A)+human MLKL (residues 190-471) complex structure. For comparative structure analysis, it would be better if author should have high resolution structure of Human RIPK3-GSK'843 complex

We appreciate the reviewer's point and agree a higher resolution structure of human RIPK3 is highly desirable. However, this is not a trivial endeavour, because the human RIPK3 kinase domain is not an especially compliant protein owing to its low thermal stability (which we now emphasize on page 5, lines 104-105). It is noteworthy that this will be the first report of the human RIPK3 kinase domain structure, despite its implication in the necroptosis pathway 12 years ago, which really speaks to the challenges associated with this protein.

Abstract section:

Page 2: lines 15-20 contains only background and rationale behind current study, while lines 20-25 contains very little information about results and significance of current work. The authors should outline clearly critical findings and their implications discovered from current work in the Abstract section.

We completely understand the reviewer's point, however the *Nature Communications* author guidelines severely restrict the amount of detail we can provide in the Abstract with the limit of 150 words. With the reviewer's comment in mind we have reworded parts of the abstract to provide further detail of the findings.

Page 2, lines 16-17: How pathway involved in dysregulation of necroptosis in inflammatory diseases could offer as key therapeutic target?

We apologize if this was unclear. We have reworded this statement for clarity; our reasoning is further outlined in the Introduction, where word count restrictions do not limit the detail we can provide.

Introduction section:

Page 4, lines 69-70: ... "RIPK3 and MLKL bound in a face-to-face arrangement with apposing activation loops" ..., Please rewrite the current sentence and explain clearly the meaning of face to face arrangement.

We have now revised on page 4, lines 74-76 to make clear that we mean the activation loops and ATP binding sites are facing one another in this arrangement, which arises from N-lobe:N-lobe and C-lobe:C-lobe contacts. We feel that this description should now be unambiguous.

Page 4, lines 76-77: Meaning of the paragraph starting with " The activation loop..... RIPK3 N-lobe binding" is not clear and should be rewritten.

We thank the reviewer for identifying this part of the Introduction as unclear. We have now rewritten this section for improved clarity on page 4, lines 81-86.

Page 4, lines 78-79: Please include briefly here the critical finding obtained from mutational analysis of interface residues within the C-lobes of the mouse RIPK3-MLKL complex, if available.

We thank the reviewer for this suggestion. We have amended this section to include greater detail (page 4-5, lines 86-91).

Results section:

Page 5, lines 86-88: Why individually purified human RIPK3 and human MLKL did not assemble as complex, when expressed individually and mixing them together, as observed in co-expression?

-Did authors got some clues in comparative structure analysis between human RIPK3 with human RIPK3+MLKL complex structures

This is an excellent question, and this remains a subject that we are actively pursuing. At this point, the basis for complex pre-assembly on co-expression is not clear to us, but our data clearly point to human RIPK3 S227 phosphorylation as a crucial determinant of MLKL engagement (new **Figure 5a**). Why the two proteins do not form a stable complex when prepared individually and mixed is currently unclear; we are investigating whether this could be dictated by additional phosphorylation events or whether phosphatase activity towards the isolated human RIPK3 kinase domain might eliminate the key pS227 site. We expect this work to form the basis of another report in the future.

*Page 5, lines 92-94: What is the intrinsic instability in human RIPK3, which prevented the human RIPK3:MLKL complex crystallization?
-How compound-10 helped in binary complex formation?*

We apologize that our reasoning was not clear in our first submission. We have now elaborated in the revised manuscript (page 5, lines 104-106) to further describe our thermal stability data in Suppl. Figure 1a, which indicate that human RIPK3 is starting to unfold at room temperature, and indeed precipitates on standing at room temperature, while addition of Compound-10 markedly enhanced thermal stability. On the basis of the stability conferred by Compound-10 binding in these experiments, we concluded that compound binding likely stabilizes the human RIPK3:MLKL complex, which enhanced our capacity to crystallize the complex. It is important to note, however, that Compound-10 does not aid complex formation; our SAXS data indicate that the complex forms with or without compound present and the complex architecture is grossly comparable with or without compound.

Page 5, lines 106-108: What could be the possible role of T224 phosphorylation in RIPK3, as S227 phosphorylation is designated as determinant of MLKL interaction.

We thank the reviewer for prompting further elaboration on this point. We have posed that the role of pT224 in RIPK3 may be an auxiliary function, rather than an obligate MLKL binding role (pages 5-6, lines 120-121). Please bear in mind that our intention is to describe its occurrence in the structure in this section, and it is further examined functionally later in the manuscript.

Page 6, lines 113-114: Compound-10 is only specific for RIPK3 ATPase pocket. How high concentration of compound-10 promote binding to MLKL ATPase pocket? Do both proteins have similar ATPase pockets?

The protein complex was pre-incubated with approximately 400 μM Compound-10 for crystallization and 40 μM Compound-10 for thermal stability assays. These are four or more orders of magnitudes higher than reported IC_{50} of 0.0029 μM towards RIPK3 (Hart *et al.*, ref. 47). Furthermore, Compound-10 is known to be promiscuous; it was originally described as a c-Met kinase inhibitor (Hart *et al.*, ref. 47). c-Met kinase, RIPK3 and MLKL contain very different ATP-binding pockets, which is reflected in the unconventional MLKL-binding mode where only half of the molecule occupies the ATP-binding pocket. Therefore, we concluded that MLKL binding likely arises as a result of the very high concentration of the compound in the crystal drop, and we do not believe this is biologically relevant. We have now clarified this important point on page 6, lines 138-139, in light of the reviewer's query.

Page 6, lines 137-138: Did authors observed any change in globular shape of SAXS envelope of human RIPK3:MLKL complex in presence and absence of compound-10 ?

-If not, is it due to fact they used higher concentration of Glycerol in SAXS buffer?

The reviewer is correct; we did not observe any differences in globular shape in SAXS experiments performed on the RIPK3:MLKL complex in the presence or absence of Compound-10. The overlap of the SAXS profiles illustrate that the globular shapes are almost identical with or without compound. We have not performed *ab initio* modelling, however, because this is simply a different presentation of our CRYSOLOG analysis and, while a visual representation, is less scientifically robust.

Because of radiation damage that occurs in SAXS experiments, it is typical to include 5% glycerol in the SEC buffer to act as a radical scavenger. The buffer used in SAXS experiments is equivalent to that used in our SEC for protein purification before crystallization trials. Accordingly, we would not anticipate any impact on conformation, because the protein subjected to crystallization trials also contained 5% glycerol.

Page 7, line 167: Salt bridge and hydrophobic “regulatory” (R)-spine interactions are hallmarks of an inactive MLKL open conformation, when bound to RIPK3 and similar conformation was observed in Mb32 bound MLKL (PDB: 7JXU).

-Do the conformational changes in MLKL were identical in RIPK3 and Mb32 bound states?

We thank the reviewer for seeking clarification on this important point. Yes, these structures are almost identical (Response Figure 5) with RMSD = 0.655 over 261 C α atoms between 7MON chain A and 7JXU chain A. They share typical structural features of an open kinase conformation, including the disruption of K230-E250 salt bridge by Q356, misaligned R-spine, and the formation of activation loop helix. We have now elaborated on these features in our revised text on page 7, lines 163-167.

Page 7-8, lines 173-176:

Did structural changes in MLKL upon Phosphorylation were identical to structural changes in MLKL upon complex formation with RIPK3? that lead RIPK3:MLKL complex dissociation?

Unfortunately, we do not have a structure of phosphorylated MLKL to enable such a comparison, despite extensive efforts over the past few years. However, we have reported the structure of human MLKL bearing the phosphomimetic T357E/S358E mutations (Petrie *et al.*, *Nature Commun* 2018), which can be used as a surrogate for phosphorylated MLKL and a point of comparison for MLKL within the RIPK3:MLKL complex structure. T357E/S358E MLKL adopts a closed conformation, which differs from the open MLKL conformer observed in the RIPK3:MLKL reported in our study. With the reviewer’s point in mind, we have ensured this difference is clear in our revised text on page 15, lines 414-418, and note that the phosphorylated, closed form likely represents the form of MLKL dissociated from RIPK3.

Page 10, lines 260-261: pS227 and pT224 of RIPK3 form salt bridges with K372 and R333 of MLKL and appear integral to MLKL binding.

These salt bridges are the only key interactions in MLKL binding or other interactions also contribute in binding specificity?

The reviewer is correct that these are the key salt bridges, but there are additional hydrophobic interactions that arise from the projection of human MLKL F386 into a hydrophobic pocket in the C-lobe of human RIPK3’s kinase domain. We have amended the Figure 1 legend (line 940) and Figure 5 legend (lines 1012-1015) to ensure this is clear to the reader.

Methods section:

Page 19, line 509-510: SAXS data collected using buffer (200 mM NaCl, 20 mM HEPES pH 7.5, 5% glycerol). 5% Glycerol is high for SAXS data, as it makes buffer matching difficult and decreases the contrast? and be repeated.

As mentioned above, we typically include 5% glycerol in our buffers for SAXS acquisition owing to the requirement for a radical scavenger to guard against capillary fouling, even with buffer alone. The setup described in ref. 77 (Ryan *et al.*) that is in operation at the Australian Synchrotron delivers high flux and minimizes capillary contact with protein, and accordingly routinely delivers phenomenal signal-to-noise for >30kDa proteins >3mg/mL. As a result, any decreases in contrast that could arise due to 5% glycerol in the buffer do not measurably compromise our signal or data quality. Because we are using inline SEC for our protein injection, there is a substantial period in the SEC run before protein elutes that we can for very close buffer matching and our experience is that this leads to superior buffer subtraction relative to static data collections. In fact, we have not performed static collections in several years, because of the difficulties we experienced with buffer subtraction and the impact of small quantities of aggregates/oligomers on interpretability. We have elaborated on the approach in the Methods section to ensure it is clear to the reader on page 21, lines 587-593.

Figures legends section:

Page 35, lines 838-840, Fig. 1C legend: How human RIPK3 (grey):MLKL complex was superposed on mouse RIPK3:MLKL complex structure?

Page 37, lines 857-858, Fig 2b legend: How human MLKL (blue):Mb32 (teal) complex (PDB: 7JXU) structure was superposed with the MLKL(grey):RIPK3 (wheat) complex.

We thank the reviewer for prompting further information on the software we used. For consistency, we performed all superimposition by matching respective MLKL chain in each complex model using UCSF Chimera Matchmaker. We have updated the legends of Figures 1c and 2b to reflect this.

REVIEWER COMMENTS

Reviewer #1 (Remarks to the Author):

The authors have fully responded to my concerns. Now the manuscript will be suitable for publication in Nature Communications.

Reviewer #2 (Remarks to the Author):

The revised manuscript "Human RIPK3 maintains MLKL in an inactive conformation prior to cell death by necroptosis" maintains the original high quality and impact of its study and presentation. I support publication of the manuscript and accompanying crystal structures in their current form.

In response to my list of queries regarding the crystallography, the authors have clarified the original results in the manuscript text when appropriate, and have directly addressed many others in the rebuttal; both forms of rebuttal are sufficient in this case.

Specifically, I am satisfied with the author's treatment of the anisotropic diffraction data from the RIPK3:GSK-843 cocrystals, after very detailed further analysis presented in the rebuttal and in some cases with addendums in the main text. On this topic, the updated manuscript now includes important steps and statistics regarding data treatment in the Methods, Results, and Table 1, in a clear fashion that will allow the reader to have a more detailed description of how the structure was determined. The authors should be commended for their steadfastness in obtaining crystals and solving these challenging structures.

I note that on p. 7 of the rebuttal, the chart presenting the completeness (83%) and $I/\sigma I$ (1.9) in the shell 3.866 – 3.697 Å may represent a better resolution cutoff for comparison, as the statistics start to fall off quickly in the next shell. However, I agree that the electron density maps shown on p. 11-12 of the rebuttal and Suppl Fig 4c are sufficient for interpretation of the mainchain tracing of the structure and placement of GSK-843.

The inclusion of SAXS molecular weight determination is also a welcome and necessary addition.